# Understanding Bias Terms in Neural Representations

**Weixiang Zhang, Boxi Li, Shuzhao Xie, Chengwei Ren, Yuan Xue, Zhi Wang**[*]
Shenzhen International Graduate School, Tsinghua University
`zwx.healthy@gmail.com, libx22@mails.tsinghua.edu.cn`
`wangzhi@sz.tsinghua.edu.cn`

## Abstract

In this paper, we examine the impact and significance of bias terms in Implicit Neural Representations (INRs). While bias terms are known to enhance nonlinear capacity by shifting activations in typical neural networks, we discover their functionality differs markedly in neural representation networks. Our analysis reveals that INR performance neither scales with increased number of bias terms nor shows substantial improvement through bias term gradient propagation. We demonstrate that bias terms in INRs primarily serve to eliminate *spatial aliasing* caused by symmetry from both coordinates and activation functions, with input-layer bias terms yielding the most significant benefits. These findings challenge the conventional practice of implementing full-bias INR architecture. We propose using freezing bias terms exclusively in input layers, which consistently outperforms fully biased networks in signal fitting tasks. Furthermore, we introduce Feature-Biased INRs (Feat-Bias), which initialize input-layer bias with high-level features extracted from pre-trained models. This feature-biasing approach effectively addresses the limited performance in INR post-processing tasks due to neural parameter uninterpretability, achieving superior accuracy while reducing parameter count and improving reconstruction quality. Our code is available at *this link*.

## 1 Introduction

Implicit Neural Representation (INR) [48, 45, 30] represents a novel paradigm in signal representation, utilizing Multilayer Perceptrons (MLPs) to establish continuous mappings from spatial coordinates (e.g., $\langle x, y \rangle$ for images) to their corresponding attributes (e.g., RGB values of pixels). Despite this methodology enables compact and resolution-agnostic representation of diverse natural signals [2, 45, 5, 35, 30, 18], traditional ReLU-based networks suffer from spectral bias [37], which impedes the fitting of high-frequency signal components during INR encoding. Various approaches have been proposed to address this limitation [48, 45, 42, 21, 60, 61], with the SIREN [45] and its variants [27] emerging as the most widely adopted solutions. These networks implement periodic activation functions, such as $\sin(x)$ and $\sin((|x| + 1)x)$, aligning with formulation of Fourier series [3] and achieving state-of-the-art performance in various signal reconstruction tasks. Leveraging these design principles, INRs have demonstrated significant potential, particularly in medical imaging [31, 52, 51], data compression [11, 13, 46], and inverse problem solving [30, 25, 40].

Although periodic activation functions in INRs have demonstrated exceptional interpretability and performance, the role of **bias terms** (i.e., trainable constants added to the weighted sum of inputs for a neuro) in such special networks remains unexplored. In traditional deep learning practices, it is acknowledged that bias terms can enhance model performance by enabling activation function shifts [16], and empirical studies indicate that they significantly enhance classification accuracy, often serving as crucial contributors to final logits [49]. Current research in INRs [45, 27, 17, 15] adheres to this conventional design, incorporating bias terms in each MLP layer. However, upon closer

---

[*]Corresponding author.

39th Conference on Neural Information Processing Systems (NeurIPS 2025).

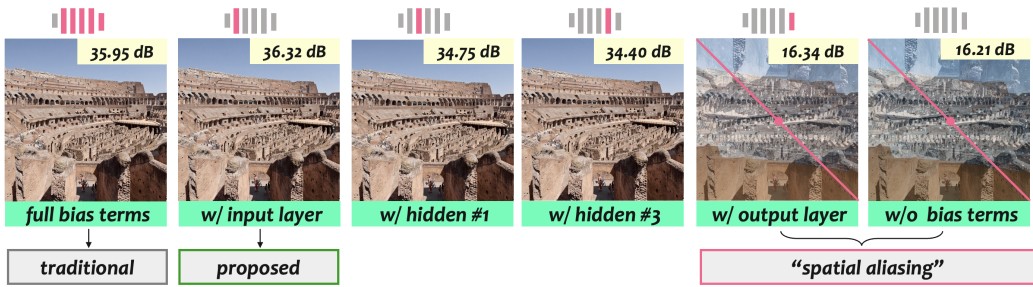

Figure 1: Impact of bias terms in INRs. Pink and Ashen in the top row indicate layers with and without bias terms, respectively. Applying bias terms (without gradient propagation[2]) to the input layer achieves optimal performance, while lacking bias terms or only applying them to output layer results in *"spatial aliasing"* manifested as central symmetry in failed reconstructions.

examination of periodic activation functions, the shifting capacity of bias terms should intuitively be significantly diminished due to the boundedness and periodicity of sine functions, in contrast to the linear growth of ReLU activation. In response to it, we propose to critically reassess bias terms in INRs by examining two questions: (1) *How significant is the impact of bias terms on INRs?* and (2) *What unique functions do bias terms serve in INRs?*

To answer these questions, we conduct empirical studies and make several findings. Consistent with intuition, bias terms $\mathbf{b}$ have significantly smaller effects than weight parameters $\mathbf{W}$, and increasing their magnitude does not consistently enhance performance. However, we unexpectedly find that optimizing bias terms through gradient descent has negligible impact on reconstruction quality. Instead, the mere presence of properly distributed bias terms proves enough critical, particularly benefiting the input layer. Based on these observations, we uncover that the primary role of bias terms in INRs is not to shift activation functions as in traditional MLPs, but rather to ***eliminate spatial aliasing caused by symmetry from both coordinates and activations***. Without bias modulation, input coordinates symmetric about the center (e.g., $\langle m, n \rangle$ and $\langle -m, -n \rangle$) would produce identical outputs, leading to failed reconstructions with central symmetry (see the right part of Fig. 1), a phenomenon we term *spatial aliasing*. Since this symmetry issue originates in coordinate space, bias terms in the input layer yield the most significant benefits. Moreover, given that bias terms have limited impact on shifting activation functions in periodically activated INRs, their presence in non-input layers, along with gradient propagation, proves negligible or even detrimental to INR encoding.

Based on these findings, we challenge the conventional practice of full-bias architecture in INR signal fitting, demonstrating that applying frozen bias terms exclusively to input layers consistently outperforms fully biased networks, achieving optimal performance relative to parameter count. Beyond enhancing INR's expressive capability, our analysis reveals an opportunity to improve *INR post-processing*, i.e., developing neural networks to process INR parameters [62, 23, 19, 15]. While this emerging task has gained importance with the proliferation of INR-represented data (also known as functa [12]), its performance remains limited by the inherent interpretability challenges of neural parameters. For instance, even the recently proposed end-to-end INR classification framework [15] achieves only approximately 60% accuracy on the CIFAR-10 dataset. Unlike existing approaches that focus on parsing intractable neural parameters [62, 23, 19] or designing task-specific end-to-end frameworks [15], we propose Featuring INRs with Bias Terms (Feat-Bias), which directly embeds high-level features from pre-trained encoders (e.g., ViTs [10, 34]) into input-layer bias space, maintaining these values throughout INR encoding. This design is motivated by our key findings that bias term gradient propagation does not affect INR reconstruction quality, thus making bias terms ideal candidates for storing well-established features. Through this approach, we achieve substantial performance improvements using only a lightweight MLP as the neural parameter processor, offering an elegant solution that simultaneously enhances reconstruction quality, reduces model parameters, and improves INR post-processing performance.

---

[2]Bias terms with gradient propagation yields nearly identical results ($\pm 0.01$dB difference for each case).

## 2 Impact of Bias Terms in INRs

### 2.1 Intuitive Analysis

Given a typical $l$-layer multilayer perceptron, the output value of the $l$-th layer can be formulated as:

$$\mathbf{z}_l = f(\mathbf{z}_{l-1}) = \sigma(\mathbf{W}_l \mathbf{z}_{l-1} + \mathbf{b}_l), \tag{1}$$

where $\mathbf{W}_l$ and $\mathbf{b}_l$ denote the weights and bias at layer $l$, handling linear transformation from the previous layer's output $\mathbf{z}_{l-1}$, and $\sigma(\cdot)$ represents the nonlinear activation function. In common practice, piecewise linear functions such as ReLU are employed as activation functions, owing to their computational efficiency and proven effectiveness across various learning-based models. However, such networks demonstrate limited performance when applied to Implicit Neural Representations (INRs), which aims to represent natural signals through coordinate-based neural networks. Formally, an INR can be defined as $F_\theta : \mathbf{x} \in \mathbb{R}^i \mapsto \mathbf{y} \in \mathbb{R}^o$, mapping an $i$-dimensional coordinate $\mathbf{x}$ to a $o$-dimensional signal $\mathbf{y}$. Given a signal set $\mathcal{S} = \{\mathbf{x_i}, \mathbf{y_i}\}_{i=1}^N$ comprising $N$ pairs of coordinates $\mathbf{x_i}$ and their corresponding signal values $\mathbf{y_i}$, the objective is to fit an $L$-layer MLP $F_\theta(\mathbf{x})$ to the ground truth $\mathbf{y}$ with minimal loss.

In this context, ReLU-based MLPs often yield suboptimal results due to *spectral bias* [37], characterized by preferential fitting of low-frequency over high-frequency components. To address this limitation, periodically activated representation network [45, 27] employ sine-based activation functions, aligning with Fourier-based signal decomposition [3] and demonstrates superior performance across various signal fitting tasks [61]. In these architectures, Eq. 1 can be rewritten as $\mathbf{z}_l = f(\mathbf{z}_{l-1}) = \sin\left(\omega_0 \alpha^l (\mathbf{W}_l \mathbf{z}_{l-1} + \mathbf{b}_l)\right)$, where $\omega_0$ controls the network frequency, and $\alpha^l$ determines the activation function's periodicity. SIREN [45] sets $\alpha^l = 1$ for a static period of $2\pi$, while FINER [27] employs $\alpha^l = |\mathbf{W}_l \mathbf{z_{l-1}} + \mathbf{b}_l| + 1$.

Intuitively, the effect of bias terms differs significantly between ReLU and sine activations. In ReLU-based networks, bias terms can provide more direct and substantial shifting of activated values, thus exerting greater influence on the network's nonlinear behavior. We can define $\delta(x, b) = \sigma(x + b) - \sigma(x)$ as an indicator to represent the effect of bias terms on activation function. For ReLU activation, where $\sigma(x) = \max(0, x)$, we obtain $\delta(x, b) = \max(0, x + b) - \max(0, x)$. Its value is bounded by $b$: $|\delta(x, b)| \leq |b|$, achieving this bound under the condition $x \geq 0 \cap b \geq -x$. In networks with periodic activation, where $\sigma(x) = \sin(\omega \alpha(x))$, we derive:

$$\delta(x, b) = 2\cos(\omega\alpha(x + \frac{2}{b}))\sin(\frac{2\omega\alpha}{b}), \tag{2}$$

with $|\delta(x, b)| \leq 2\sin(\frac{2\omega\alpha}{b}) \leq 2$. Comparing $\delta(x, b)$ between these activations reveals that bias terms provide unbounded control in ReLU networks, whereas in periodically activated networks, their modulation capability is inherently bounded by the sine function, constraining their effect on nonlinear modeling.

Except for unbounded magnitude, bias terms in ReLU activation provide stronger threshold control capabilities, a feature absent in sine activation. With ReLU activation $\sigma(x + b) = \max(0, x + b)$, neurons activate ($\sigma(x + b) > 0$) when $x > -b$, where bias $b$ directly modulates this activation threshold. Minor adjustments in $b$ can toggle neurons between active and inactive states, establishing an effective *gating mechanism*. In contrast, bias terms in sine activation merely induce *phase shifts* rather than implementing hard gating states. Furthermore, the periodic nature of sine functions constrains the effect of bias terms to a $2\pi$ period, further diminishing their contribution to the network's expressiveness.

Based on the preceding analysis, we can hypothesize that the shifting effect of bias terms in INRs is significantly diminished. To validate this hypothesis and understand its implications for signal fitting tasks, comprehensive empirical studies are necessary. Therefore, we propose to critically reassess bias terms in INRs by examining two fundamental questions: (1) *How significant is the impact of bias terms on INRs?* and (2) *What unique functions do bias terms serve in INRs?* These questions will be investigated in Sec. 2.2 and Sec. 2.3, respectively.

### 2.2 Empirical Studies

To validate our analysis from Sec. 2 regarding bias terms' impact in INRs, we conduct empirical studies under various bias-related configurations. Following prior work [41, 27, 61, 48], we implement

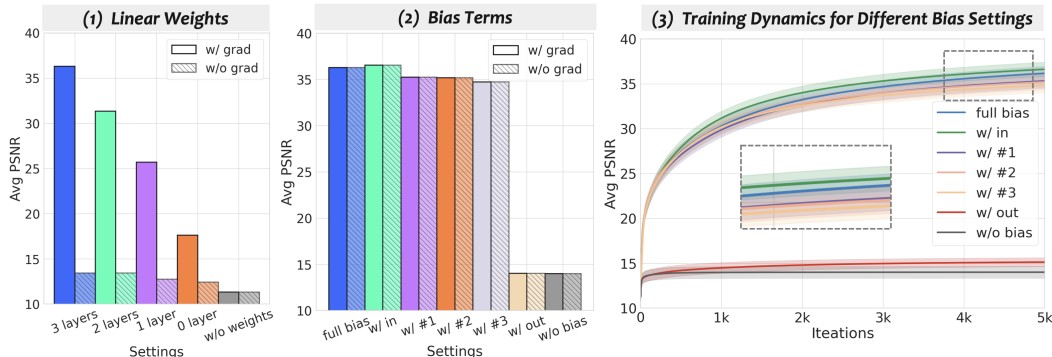

Figure 2: Left: (1) Comparative analysis of INR performance with varying numbers of weights; Middle: (2) Empirical studies on the impact of bias terms in INRs. Right: (3) Training dynamics under different bias configurations. Results demonstrate that applying bias terms exclusively to input layers yields optimal performance.

2D image fitting tasks on the Div2K dataset [1] using a $3 \times 256$ SIREN network (detailed settings and alternative backbones are provided in the appendix).

We first examine whether model performance scales with the number of bias terms. For comparison, we conduct parallel studies with varying configurations of weights $\mathbf{W}_l$ across different layers. For weights, we systematically retain different numbers of layers to investigate their effects (Fig. 2-(1)). For bias, we selectively preserve bias in specific layers while eliminating it in others for finer-grained control (Fig.2-(2)). Results shown in Fig. 2 demonstrate that weights are significantly more crucial for network performance, exhibiting a strong linear relationship between parameter count and reconstruction quality. In contrast, bias terms show minimal impact; even reducing bias terms to approximately 25% of their original number (by retaining only one layer's bias) maintains nearly consistent reconstruction quality, supporting our intuitive analysis:

**(Observation #1)** *Compared to weights, bias terms exhibit substantially smaller effects on INRs' performance, and increasing their magnitude does not consistently lead to performance improvements.*

To investigate bias effects further, we conduct experiments by blocking gradient backpropagation. Fig. 2-(1) shows that weight optimization is essential for signal reconstruction, as networks fail without their gradients. In contrast, we observe an unexpected phenomenon for bias terms:

**(Observation #2)** *The gradient optimization of bias has negligible impact on INRs.*

Our observations reveal that the impact of bias terms on INRs is even more limited than initially hypothesized, with their gradient updates having minimal effect on reconstruction quality. This suggests structural redundancy in current INR architectures, where computational resources are inefficiently allocated to storing and updating largely ineffective bias parameters. However, completely eliminating bias terms or restricting them to the output layer significantly degrades performance, indicating that strategic bias placement is more fundamental than optimization. Through comprehensive analysis of bias configurations, we discovered that bias terms in the input layer achieve optimal performance relative to parameter count. Notably, architectures with frozen input-layer bias consistently outperform fully biased configurations. In summary:

**(Observation #3)** *The existence of bias terms plays a more essential role than their optimization, with neurons in the input layer benefiting most significantly from this property.*

### 2.3 Bias can Eliminate *Spatial Aliasing*

Although empirical studies in Sec. 2.2 demonstrate that bias terms have minimal yet indispensable effects on INRs, the underlying mechanism remains unclear: *What unique functions do bias terms serve in INRs?* As revealed in Observation #2, the negligible impact of bias term optimization suggests that their conventional role in enhancing nonlinear representational capacity through activation value shifting, fundamental to ReLU-based networks, no longer applies to INRs. Upon examining reconstruction failures without bias modulation, we observe a consistent phenomenon:

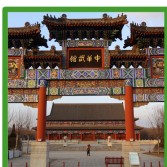 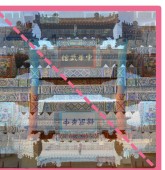 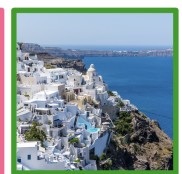 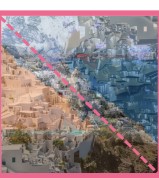 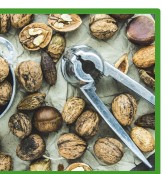 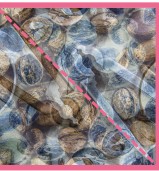

Figure 3: Visualization for *spatial aliasing*: reconstructed signals demonstrate central symmetry, manifesting as aliasing artifacts between distinct regions of the input. Green and Pink denote the ground truth and reconstructed signals with *spatial aliasing*, respectively.

pixel intensities in the reconstructed image exhibit symmetry around the image center, resulting in aliasing effects between different regions of the input image, which we term *spatial aliasing*. We propose that the primary impact of bias terms in INRs is to eliminate this *spatial aliasing*.

**Why does *spatial aliasing* happen in bias-free INRs?** We attribute this phenomenon to the combined effect of activation function symmetry and input coordinate space structure. In INR training, it is common practice to rescale the coordinate $\mathbf{x}$ to $\tilde{\mathbf{x}}$ as: $\tilde{\mathbf{x}} = \frac{2\mathbf{x}}{\max(\mathbf{x}) - \min(\mathbf{x})} - \frac{1}{2}$, which normalizes $\tilde{\mathbf{x}}$ to $[-1, 1]$. This normalization has been proven optimal for INR overfitting [45, 60]. Under this configuration, coordinates that are symmetric about the image center generate values of equal magnitude but opposite signs, which, in the absence of bias modulation, directly induces spatial aliasing. Specifically, consider the input layer of a bias-free INR with centrally symmetric coordinates $\langle m, n \rangle$ and $\langle -m, -n \rangle$, where $m, n \in [-1, 1] \setminus \{0\}$. The activated value of the $i$-th neuron can be formalized as

$$z_0^i = f(m, n) = \sin(\omega_0 \alpha^0 (W_i^0 m + W_i^1 n)). \tag{3}$$

This function exhibits central symmetry, satisfying $f(-m, -n) = -f(m, n)$, a property that extends to all neurons in the bias-free network. While removing symmetry from either the coordinate space or activation function would eliminate spatial aliasing, it would significantly degrade reconstruction quality compared to the default symmetric configuration (see Sec. 4.3). In summary:

**Proposition 1** *The primary role of bias in periodically activated INRs serves as to eliminate spatial aliasing resulted from the the symmtery of both coordinate space and activation function, with neglectable effect on enhance the nonlinear expressiveness of model with activation shifting.*

**Discussion.** The distinct role of bias terms in INRs, as established in Proposition 1, aligns with both our intuitive analysis (Sec. 2) and empirical observations(Sec. 2.2). This finding provides additional explanations for the training dynamics under various bias configurations in Fig. 2-(3): **(1)** Bias terms achieve maximal effectiveness in the input layer due to direct coordinate processing. While hidden layers can also mitigate spatial aliasing, their efficacy gradually diminishes with increasing depth from the input layer. **(2)** The output layer's limited neurons (matching the output signal dimension, three for images) provide only marginal improvement over bias-free networks. This minimal bias modulation proves insufficient to fully address spatial aliasing due to the extremely constrained number of bias. **(3)** More bias terms do not necessarily enhance performance: INRs with bias terms exclusively in the input layer outperform fully-biased configurations. This suggests that while input layer bias effectively eliminates spatial aliasing, bias terms in deeper layers provide negligible activation shifting and potentially act as low-magnitude noise, slightly compromising network overfitting capacity.

## 3 Application: Featuring INRs with Bias Terms

In this section, we introduce Featuring INRs with Bias (Feat-Bias), a method that leverages our analysis of bias terms to enhance INR post-processing tasks, which aim to develop neural networks for processing INR parameters (e.g., INR classification task) [62, 23, 19, 15]. While this emerging field has gained significance with the proliferation of INR-represented data (also known as functa [12]), performance remains limited due to the inherent interpretability challenges of neural parameters. For instance, even the recently proposed end-to-end INR classification framework [15] achieves only approximately 60% accuracy on the CIFAR-10 dataset. Current methods aiming on INR post-processing broadly fall into two categories: symmetry-equivariant parsers for neural parameters and task-specific end-to-end frameworks. The former focuses on designing parameter parsers that

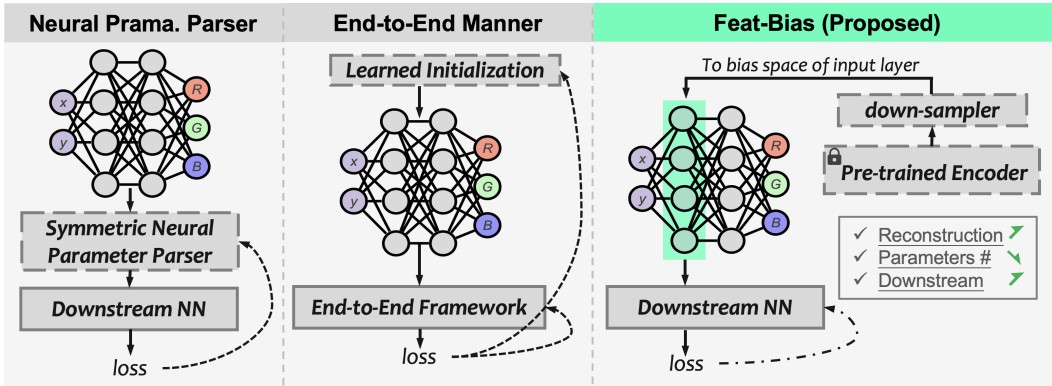

Figure 4: Comparison of mechanisms in INR post-postprocessing. Left: Neural parameter parsers that address the inherent permutation symmetries of neurons. Middle: End-to-end frameworks that integrate INR encoding directly into task-specific networks. Right: Feat-Bias (proposed) that embeds high-level features into input-layer bias initialization. Our method leverages the understanding of bias terms in INRs, addressing neural parameter complexity through a straightforward approach that surpasses existing methods.

consider the inherent permutation symmetries of neurons, while the latter integrates INR encoding directly into task-specific networks, demonstrating rather promising results.

Our Feat-Bias method differs by directly embedding high-level features into neural parameters (specifically bias terms of the input layer) at the initialization step, maintaining these values fixed throughout overfitting. This design is motivated by our empirical observations and Proposition 1: since the backpropagation of bias terms in INRs does not affect reconstruction, these values remain constant throughout the INR encoding process. This stability makes bias terms ideal storage spaces for high-level features extracted from well-established pre-trained encoders (e.g., ViTs [10, 34]). By utilizing a lightweight MLP to process these bias values, we achieve excellent performance while circumventing the challenging process of extracting features from intractable weights.

Specifically, we find that feature vectors extracted from pre-trained encoders approximately follow a uniform distribution. By rescaling these features to $[-\frac{1}{\sqrt{n}}, \frac{1}{\sqrt{n}}]$ (where $n$ denotes the number of input neurons) to match default bias initialization, we maintain reconstruction performance while significantly enhancing INRs' feature representation capabilities. Moreover, while INR input layer dimensions can be flexibly adjusted, pre-trained models produce fixed-dimension features (e.g., 768 dimensions for ViT-Base). This dimensional mismatch can be efficiently resolved through linear interpolation. Due to the high expressiveness of the encoded features, we achieve state-of-the-art performance in INR post-processing tasks even with significant dimensionality reduction. While Feat-Bias offers limited insights into permutation-symmetric neural parameter processing, it demonstrates a practical application of our understanding of bias terms in neural representations, providing an efficient solution for INR post-processing.

# 4 Experiments

## 4.1 INR Representation Task

**Experimental Settings.** In this section, we demonstrate how our findings regarding bias terms can enhance INR representation performance,serving as a detailed extension of our empirical studies (Sec. 2.2). Following prior work [41, 27, 61, 48], we conduct 2D image fitting tasks on the DIV2K dataset [1] (additional datasets are detailed in supplementary materials). We employ a $3\times256$ MLP with SIREN [45] and FINER [27] architectures, setting the total number of iterations $T$ to 5000. The reconstruction quality is evaluated using PSNR, SSIM [50], and LPIPS [58] metrics. All experiments in this section were performed using the PyTorch framework [36] on NVIDIA RTX 3090 GPU with 24.58 GB VRAM.

**Quantitative Results.** The quantitative results are presented in Tab. 1. Building upon our findings in Sec. 2.2, we challenge the traditional INR architecture that applies bias terms in all layers (Standard) by proposing a model that utilizes bias terms solely in the input layer (Input Bias). We evaluate this against alternative bias configurations: a network without any bias terms (Bias Free), and networks with bias terms only in the first hidden layer ($H_1$ Bias) or output layer (Output Bias). Except for the Standard configuration, all variants operate without bias term backpropagation, as gradient propagation through bias terms yields negligible differences ($\pm$ 0.01dB for each case). As shown in Tab. 1, our proposed Input Bias consistently enhances reconstruction quality across both SIREN and FINER architectures, achieving maximum PSNR improvements of 0.88 dB and 1.13 dB, respectively. The Bias Free and Output Bias configurations exhibit spatial aliasing artifacts due to the absence of appropriate bias terms, confirming our analysis in Sec. 2.3. Furthermore, our mechanism not only improves performance but also reduces the parameter count of each INR by approximately $\frac{(l-1)n}{ln^2+ln}$, where $l$ denotes the number of hidden layers and $n$ represents the neuron count per hidden layer.

Table 1: Results of 2D image fitting tasks with varying bias configurations[3].

| Settings | 1k Iterations | | 3k Iterations | | 5k Iterations | | |
|---|---|---|---|---|---|---|---|
| | PSNR $\uparrow$ | SSIM$\uparrow$ | PSNR$\uparrow$ | SSIM$\uparrow$ | PSNR$\uparrow$ | SSIM$\uparrow$ | LPIPS$\downarrow$ |
| Standard | $30.30_{\pm0.05}$ | $0.881_{\pm0.002}$ | $34.69_{\pm0.07}$ | $0.948_{\pm0.001}$ | $36.19_{\pm0.05}$ | $0.959_{\pm0.001}$ | $0.023_{\pm0.001}$ |
| Bias Free$^\oslash$ | $13.98_{\pm0.00}$ | $0.556_{\pm0.000}$ | $14.00_{\pm0.00}$ | $0.574_{\pm0.001}$ | $14.03_{\pm0.01}$ | $0.577_{\pm0.000}$ | $0.415_{\pm0.000}$ |
| $H_1$ Bias | $29.87_{\pm0.02}$ | $0.870_{\pm0.001}$ | $34.00_{\pm0.03}$ | $0.938_{\pm0.001}$ | $35.34_{\pm0.04}$ | $0.950_{\pm0.001}$ | $0.035_{\pm0.001}$ |
| Output Bias$^\oslash$ | $14.49_{\pm0.01}$ | $0.562_{\pm0.001}$ | $14.96_{\pm0.01}$ | $0.583_{\pm0.000}$ | $15.13_{\pm0.01}$ | $0.588_{\pm0.000}$ | $0.407_{\pm0.002}$ |
| Input Bias | $31.18_{\pm0.01}$ | $0.902_{\pm0.001}$ | $35.32_{\pm0.01}$ | $0.953_{\pm0.002}$ | $36.63_{\pm0.01}$ | $0.962_{\pm0.00}$ | $0.019_{\pm0.001}$ |
| Standard | $32.25_{\pm0.09}$ | $0.916_{\pm0.002}$ | $36.83_{\pm0.07}$ | $0.962_{\pm0.001}$ | $38.47_{\pm0.05}$ | $0.971_{\pm0.001}$ | $0.014_{\pm0.001}$ |
| Bias Free$^\oslash$ | $13.99_{\pm0.01}$ | $0.568_{\pm0.000}$ | $14.01_{\pm0.00}$ | $0.581_{\pm0.003}$ | $14.03_{\pm0.01}$ | $0.583_{\pm0.001}$ | $0.415_{\pm0.000}$ |
| $H_1$ Bias | $31.08_{\pm0.03}$ | $0.894_{\pm0.001}$ | $35.23_{\pm0.01}$ | $0.949_{\pm0.000}$ | $36.62_{\pm0.02}$ | $0.958_{\pm0.001}$ | $0.026_{\pm0.001}$ |
| Output Bias$^\oslash$ | $14.51_{\pm0.01}$ | $0.573_{\pm0.000}$ | $14.98_{\pm0.01}$ | $0.590_{\pm0.000}$ | $15.14_{\pm0.00}$ | $0.594_{\pm0.000}$ | $0.406_{\pm0.001}$ |
| Input Bias | $33.38_{\pm0.06}$ | $0.932_{\pm0.001}$ | $37.90_{\pm0.03}$ | $0.968_{\pm0.001}$ | $39.29_{\pm0.02}$ | $0.974_{\pm0.000}$ | $0.010_{\pm0.000}$ |

## 4.2 INR Classification Task

**Experimental Settings.** In this section, we evaluate the effectiveness of our proposed Feat-Bias (Sec. 3) on INR downstream tasks. Following prior work [62, 19, 15], we focus on INR classification as the primary task. We compare Feat-Bias against state-of-the-art methods including NFN [62], ScaleGMN [19], Zoo$_{[ViT]}$ [29], and MWT [15], encompassing both symmetry-equivariant neural parameter parsers and end-to-end approaches. Due to code availability constraints, results marked with † are quoted from original papers. The downstream network of Feat-Bias consists of a lightweight $3 \times 256$ MLP classifier, trained for 1000 iterations using a cosine scheduler with a learning rate of $1e-3$. All experiments are repeated three times, reporting both mean and standard deviation. Additional experiments regarding this section can be found in supplementary material.

**Dataset of Neural Representations.** Beyond standard image datasets, establishing a large-scale dataset of neural representations is crucial. While Implicit-Zoo [29] provides CIFAR-10 in neural representation form, their architecture is incompatible with Feat-Bias, which requires specific bias initialization in the input layer. Therefore, we re-train the CIFAR-10 neural representation dataset using a $1 \times 64$ SIREN network trained for 1000 epochs, maintaining alignment with Implicit-Zoo [29] parameters except for the bias initialization scheme. Our implementation achieves an average reconstruction PSNR of 38dB across the CIFAR-10 dataset, with an average training time of 3.11 seconds per image versus Implicit-Zoo's reported 10 seconds.

**Quantitative Results.** Tab. 2 shows the classification results for CIFAR-10 dataset [24], where we adopt Implicit-Zoo's default configuration with a $1 \times 64$ SIREN network to ensure consistent experimental conditions across all comparisons. For all baselines except end-to-end MWT [15] and our method, we directly implement the released checkpoints from Implicit-Zoo as input to the neural parameter parser network. As evidenced by the results, our method achieves superior performance across accuracy, precision, and F1 score metrics, demonstrating Feat-Bias's effectiveness in INR post-processing tasks. While our reported performance is lower than the original MWT paper due to

---

[3]Settings without underlines represent standard SIREN implementations, while underlined settings correspond to FINER; $\oslash$ represents reconstructions exhibiting *spatial aliasing*.

utilizing smaller network architecture ($1 \times 64$ versus their $3 \times 256$), it is noteworthy that even their state-of-the-art results on CIFAR-10 classification peaked at 60% accuracy, substantially below the performance of our method. Additionally, our approach maintains high reconstruction quality without the accuracy-reconstruction trade-off commonly observed in end-to-end methods. Furthermore, by implementing a lightweight MLP processor, we significantly reduce both computational overhead and parameter count, bringing execution time from minutes (min) to seconds (sec) [4].

Table 2: INR classification on CIFAR-10 Datasets

| Method | Classification Task | | | | | INRs | |
| | Accuracy (%) ↑ | Precision (%) ↑ | F1 ↑ | Time ↓ | Params (kilo #) ↓ | PSNR (dB) ↑ | SSIM ↑ |
|---|---|---|---|---|---|---|---|
| NFN$_{NP}$ [62] | $26.19_{\pm0.06}$ | $24.62_{\pm1.83}$ | $23.30_{\pm1.39}$ | $25.02_{\pm0.98}$ (min) | 544 | 34.59 | 0.968 |
| NFN$_{HNP}$ [62] | $25.61_{\pm0.67}$ | $28.10_{\pm1.39}$ | $23.89_{\pm0.65}$ | $23.01_{\pm0.94}$ (min) | 1672 | 34.59 | 0.968 |
| ScaleGMN [19] | $55.31_{\pm0.64}$ | $56.10_{\pm1.33}$ | $55.25_{\pm0.95}$ | $215.86_{\pm0.28}$ (min) | 397 | 34.59 | 0.968 |
| ScaleGMN-B [19] | $56.14_{\pm0.80}$ | $57.67_{\pm0.53}$ | $56.12_{\pm0.79}$ | $256.32_{\pm0.99}$ (min) | 492 | 34.59 | 0.968 |
| WT [15] | $39.34_{\pm2.13}$ | $38.55_{\pm2.05}$ | $38.47_{\pm2.15}$ | $71.75_{\pm1.25}$ (min) | 261 | 29.08 | 0.887 |
| MWT$_{Mid-Task}$[‡] [15] | $43.38_{\pm1.35}$ | $42.90_{\pm1.43}$ | $42.95_{\pm1.42}$ | $79.74_{\pm0.98}$ (min) | 261 | 27.53 | 0.868 |
| MWT [15] | $46.94_{\pm0.37}$ | $46.48_{\pm0.50}$ | $46.51_{\pm0.46}$ | $79.49_{\pm2.44}$ (min) | 261 | 23.23 | 0.695 |
| Zoo$_{[ViT]}$ [29][†] | $80.82_{\pm0.86}$ | $80.76_{\pm0.87}$ | $80.75_{\pm0.86}$ | / | ∼5k | 34.59 | 0.968 |
| Zoo$_{[ViT]}$ [29][†] + S | $80.24_{\pm0.47}$ | $80.49_{\pm0.63}$ | $80.44_{\pm0.57}$ | / | ∼5k | 34.59 | 0.968 |
| Zoo$_{[ViT]}$ [29][†] + LC | $81.33_{\pm0.23}$ | $81.29_{\pm0.22}$ | $81.30_{\pm0.23}$ | / | ∼5k | 34.59 | 0.968 |
| Zoo$_{[ViT]}$ [29][†] + LP | $79.51_{\pm0.23}$ | $79.37_{\pm0.34}$ | $79.37_{\pm0.35}$ | / | ∼5k | 34.59 | 0.968 |
| Zoo$_{[ViT]}$ [29][†] + LP[*] | $81.57_{\pm0.29}$ | $81.53_{\pm0.30}$ | $81.51_{\pm0.30}$ | / | ∼5k | 34.59 | 0.968 |
| Feat-Bias$_{[ViT]}$ | $91.68_{\pm0.19}$ | $91.70_{\pm0.19}$ | $91.68_{\pm0.19}$ | $64.96_{\pm11.34}$ (sec) | 151 | 37.99 | 0.991 |
| Feat-Bias$_{[DINOV2]}$ | $93.78_{\pm0.21}$ | $93.80_{\pm0.22}$ | $93.78_{\pm0.22}$ | $64.35_{\pm10.91}$ (sec) | 151 | 38.02 | 0.991 |

[‡] denotes $\omega_{task}$ reported in MWT [15].

## 4.3 Ablation Study: Additional Verification for *Spatial Aliasing*

**Motivations and Settings.** In this section, we further investigate the source of spatial aliasing and elucidate why bias terms cannot be entirely eliminated from INR architectures. Consistent with the settings in Sec. 4.1, we conduct an ablation study examining the symmetry of coordinate space and activation functions to explore their impact on the reconstruction quality of INRs. We denote the elimination of coordinate space symmetry as w/o coord. sym., achieved by rescaling coordinates to $[0, 1]$ rather than $[-1, 1]$. Similarly, w/o act. sym. ($\pm 1$) indicates the removal of activation function symmetry by modifying the activation function from $\sin(\omega_0 \alpha(\cdot))$ to $\sin(\omega_0 \alpha(\cdot)) \pm 1$.

**Results and Analysis.** The results are presented in Tab. 3, where each block reports reconstruction metrics (PSNR / SSIM / LPIPS). Our experiments demonstrate that spatial aliasing occurs exclusively in bias-free networks that maintain symmetry in both coordinate space and activation functions, providing strong evidence that this source of spatial aliasing is unique to INR architectures. While removing symmetry from either component eliminates spatial aliasing, it significantly compromises reconstruction quality, yielding results substantially inferior to conventional practice. Therefore, employing bias terms solely in the input layer can effectively eliminate spatial aliasing while maintaining the symmetrical properties of INR, thus achieving optimal reconstruction quality.

Table 3: Ablation study for *spatial aliasing*

| Settings | standard | w/o coord. sym. | w/o act. sym. ($+1$) | w/o act. sym. ($-1$) |
|---|---|---|---|---|
| Full-Bias | 36.19 / 0.962 / 0.023 | 33.09 / 0.929 / 0.069 | 30.86 / 0.892 / 0.133 | 30.85 / 0.891 / 0.134 |
| Bias-Free | **14.03 / 0.577 / 0.415** | 33.01 / 0.928 / 0.073 | 30.51 / 0.882 / 0.148 | 30.48 / 0.882 / 0.149 |

[4]Due to the unavailability code of Zoo$_{[ViT]}$ [29], we cannot measure their exact computational metrics; however, their transformer-based INR classification network inherently requires more computational resources than our MLP-based approach.

# 5 Related Work

## 5.1 Implicit Neural Representations

Implicit Neural Representation (INR) [48, 45, 30] introduces an advanced approach that employs coordinate-based multilayer perceptrons (MLPs) for multimedia data representation and storage. The spectral bias phenomenon [37], wherein high-frequency signal components are more challenging to fit during the INR encoding process, has prompted numerous enhancement strategies. These improvements span various aspects, including novel activation function designs [45, 41, 39, 27], accelerated sampling techniques [59, 22, 57, 61], meta-learning frameworks [47, 7, 12], data transformation methods [43, 60], and alternative approaches [42, 44, 55, 32]. These advancements have enhanced the capacity of INRs to exploit their inherent memory efficiency and natural suitability for inverse problems, enabling various downstream applications including image enhancement [26, 9, 6], view synthesis [30, 4, 32, 54, 53, 56], PDE solving [38, 8, 20], data compression [13, 46, 17]. Despite extensive research in this field, the role of bias terms in INRs remains unexplored. Our work bridges this gap through comprehensive analysis and leverages these findings to enhance INRs' expressive capability and downstream performance.

## 5.2 Neural Networks for Neural Representations

Unlike discrete representations such as pixels, neural representations contain less interpretable parameters, making them challenging for designing network for processing them. Existing methods in this domain fall into two distinct categories: **(1) Symmetry-equivariant architectures** [33, 62, 23, 19]. This line of research considers that neural parameters exhibit inherent symmetries, such as permutation symmetries, where neurons can be arbitrarily permuted without affecting their behavior. The goal is to design architectures that are equivariant to these symmetries. NFN [62] proposes Equivariant NF-Layers, which separately address the assumed symmetries of hidden neurons (HNP) and all neurons (NP), demonstrating effectiveness in downstream tasks. NG-GNN [23] represents the input of INRs as a neural graph structure and leverages well-established Graph Neural Networks (GNNs) for processing. ScaleGMN [19] extends the scope of such meta-architectures from permutation symmetries to scaling symmetries and introduces a corresponding equivariant message-passing framework, achieving superior performance against other meta-network. **(2) End-to-end methods** [15]. Recently, MWT [15] was introduced to address this problem without explicitly considering the equivariant symmetries of INRs. Instead, it integrates the encoding process of INRs directly into the task-specific network's loop in an end-to-end manner, achieving state-of-the-art performance on the INR classification task. **Differences:** Unlike these approaches, Feat-Bias infuses established features into input-layer bias space. While offering limited insights into permutation-symmetric processing, it achieves significant performance improvements through a lightweight approach, leveraging novel findings about bias terms in INRs.

# 6 Conclusion and Limitations

This paper reveals the previously overlooked role of bias terms in INRs through analytical and empirical studies. Our findings show that, unlike in conventional MLPs, bias terms in INRs contribute minimally to nonlinear representation, with their gradient propagation having negligible impact on performance. Through comprehensive analysis, we demonstrate that bias terms primarily mitigate spatial aliasing caused by coordinate and activation symmetries, with input layer bias terms providing the most significant benefits. Based on these insights, we propose applying bias terms exclusively to input layers, challenging the conventional full-bias architecture and demonstrating consistent performance improvements. Moerover, we introduce Featuring INRs with Bias Terms (Feat-Bias), which initializes input-layer bias with pre-trained encoder features, bypassing the challenging process of parsing intractable neural parameters. This approach achieves substantial performance gains while reducing computational costs compared to existing INR post-processing methods.

The main limitation of our work lies in the lack of rigorous mathematical proof to theoretically explain the surprising behavior of bias terms in INRs, despite our intuitive analysis and empirical evidence. Future work will focus on developing a formal theoretical framework for these findings.

## Acknowledgments and Disclosure of Funding

This work is supported in part by National Key Research and Development Project of China (Grant No. 2023YFF0905502), National Natural Science Foundation of China(Grant No. 92467204 and 62472249), and Shenzhen Science and Technology Program (Grant No. JCYJ20220818101014030 and KJZD20240903102300001).

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

# A   Additional Verification for Empirical Studies

In this section, we provide additional verification of our empirical studies (Sec.2.2) and INR representation experiments (Sec.4.1). We conduct experiments on the Kodak dataset [14], maintaining all configurations identical to those described in Sec. 4.1. The quantitative results are presented in Tab. 4. The configurations Standard, Bias Free, $H_1$ Bias, Output Bias, and Input Bias align with those in Tab. 1. Additionally, we include $H_2$ Bias and $H_3$ Bias, representing networks with bias terms exclusively in the second and third layers, respectively. Settings without underlines represent standard SIREN implementations, while underlined settings correspond to FINER. The symbol $\oslash$ indicates reconstructions exhibiting *spatial aliasing*. For each configuration block, we report averaged metrics both with and without bias term backpropagation. Complementary results on the DIV2K dataset are presented in Tab. 5, supplementing the findings in Tab. 1. The results in Tab. 5 and Tab. 4 further validate our observations from Sec. 2.2 and conclusions from Sec. 4.1. These findings substantiate both the magnitude of bias terms' impact in INRs and the effects of their gradient optimization, lending additional support to Proposition 1. This reinforces our claim that the primary role of bias terms in neural representations is to eliminate *spatial aliasing* arising from the symmetry of coordinate space and activation functions.

Table 4: Results of 2D image fitting tasks with varying bias configurations (Kodak dataset)

| Settings | 1k Iterations PSNR ↑ | SSIM↑ | 3k Iterations PSNR↑ | SSIM↑ | 5k Iterations PSNR↑ | SSIM↑ |
|---|---|---|---|---|---|---|
| Standard | 30.05 / 30.05 | 0.832 / 0.832 | 33.93 / 33.92 | 0.912 / 0.912 | 35.24 / 35.23 | 0.929 / 0.929 |
| Bias Free$^\oslash$ | 15.06 / 15.06 | 0.599 / 0.599 | 15.09 / 15.09 | 0.620 / 0.620 | 15.10 / 15.10 | 0.626 / 0.626 |
| $H_1$ Bias | 29.67 / 29.68 | 0.822 / 0.822 | 33.25 / 33.25 | 0.902 / 0.902 | 34.44 / 34.45 | 0.919 / 0.919 |
| $H_2$ Bias | 29.96 / 29.96 | 0.831 / 0.831 | 33.33 / 33.32 | 0.903 / 0.903 | 34.41 / 34.42 | 0.918 / 0.918 |
| $H_3$ Bias | 30.07 / 30.05 | 0.829 / 0.829 | 33.17 / 33.16 | 0.895 / 0.894 | 34.18 / 33.17 | 0.910 / 0.910 |
| Output Bias$^\oslash$ | 16.34 / 16.04 | 0.613 / 0.610 | 17.03 / 16.78 | 0.641 / 0.639 | 17.31 / 17.02 | 0.648 / 0.646 |
| Input Bias | 30.78 / 30.78 | 0.854 / 0.854 | 34.31 / 34.30 | 0.917 / 0.917 | 35.42 / 35.42 | 0.930 / 0.930 |
| Standard | 32.05 / 32.06 | 0.880 / 0.881 | 36.18 / 36.18 | 0.940 / 0.940 | 37.52 / 37.53 | 0.952 / 0.952 |
| Bias Free$^\oslash$ | 15.09 / 15.09 | 0.617 / 0.617 | 15.11 / 15.11 | 0.632 / 0.632 | 15.11 / 15.11 | 0.636 / 0.636 |
| $H_1$ Bias | 30.83 / 30.82 | 0.853 / 0.853 | 34.46 / 34.45 | 0.921 / 0.921 | 35.82 / 35.81 | 0.935 / 0.935 |
| $H_2$ Bias | 31.28 / 31.28 | 0.866 / 0.866 | 34.74 / 34.75 | 0.924 / 0.924 | 35.66/ 35.67 | 0.934 / 0.935 |
| $H_3$ Bias | 31.37 / 31.36 | 0.865 / 0.864 | 34.60 / 34.60 | 0.919 / 0.919 | 35.60 / 35.59 | 0.931 / 0.931 |
| Output Bias$^\oslash$ | 16.19 / 16.07 | 0.628 / 0.627 | 16.82 / 16.79 | 0.652 / 0.651 | 17.10 / 17.04 | 0.655 / 0.657 |
| Input Bias | 33.06 / 33.05 | 0.901 / 0.900 | 36.81 / 36.80 | 0.948 / 0.950 | 37.92 / 37.91 | 0.955 / 0.955 |

Table 5: Results of 2D image fitting tasks with varying bias configurations (DIV2K dataset)

| Settings | 1k Iterations PSNR ↑ | SSIM↑ | 3k Iterations PSNR↑ | SSIM↑ | 5k Iterations PSNR↑ | SSIM↑ |
|---|---|---|---|---|---|---|
| Standard | 30.30 / 30.30 | 0.882 / 0.882 | 34.68 / 34.69 | 0.946 / 0.946 | 36.17 / 36.18 | 0.956 / 0.959 |
| Bias Free$^\oslash$ | 13.98 / 13.98 | 0.556 / 0.556 | 14.00 / 14.00 | 0.574 / 0.574 | 14.03 / 14.03 | 0.577 / 0.577 |
| $H_1$ Bias | 29.88 / 29.87 | 0.872 / 0.872 | 34.02 / 34.00 | 0.938 / 0.938 | 35.33 / 35.34 | 0.950 / 0.950 |
| $H_2$ Bias | 30.14 / 30.13 | 0.875 / 0.875 | 33.98 / 33.96 | 0.935 / 0.935 | 35.20 / 35.19 | 0.946 / 0.946 |
| $H_3$ Bias | 30.18 / 30.14 | 0.871 / 0.869 | 33.65 / 33.63 | 0.927 / 0.926 | 34.77 / 34.76 | 0.934 / 0.933 |
| Output Bias$^\oslash$ | 14.67 / 14.49 | 0.567 / 0.562 | 15.11 / 14.96 | 0.593 / 0.583 | 15.11 / 15.13 | 0.586 / 0.588 |
| Input Bias | 31.20 / 31.18 | 0.903 / 0.902 | 35.33 / 35.32 | 0.953 / 0.953 | 36.61 / 36.63 | 0.962 / 0.962 |
| Standard | 32.26 / 32.25 | 0.916 / 0.916 | 36.84 / 36.83 | 0.962 / 0.962 | 38.47 / 38.47 | 0.971 / 0.971 |
| Bias Free$^\oslash$ | 13.99 / 13.99 | 0.568 / 0.568 | 14.01 / 14.01 | 0.581 / 0.581 | 14.03 / 14.03 | 0.583 / 0.583 |
| $H_1$ Bias | 31.06 / 31.08 | 0.894 / 0.894 | 35.21 / 35.23 | 0.950 / 0.949 | 36.63 / 36.62 | 0.958 / 0.958 |
| $H_2$ Bias | 31.49 / 31.49 | 0.901 / 0.901 | 35.28 / 35.24 | 0.948 / 0.947 | 36.62 / 36.60 | 0.957 / 0.957 |
| $H_3$ Bias | 31.66 / 31.60 | 0.901 / 0.899 | 35.21 / 35.18 | 0.946 / 0.944 | 36.32 / 36.30 | 0.952 / 0.951 |
| Output Bias$^\oslash$ | 14.55 / 14.51 | 0.577 / 0.573 | 15.13 / 14.98 | 0.599 / 0.590 | 15.22 / 15.14 | 0.595 / 0.594 |
| Input Bias | 33.39 / 33.38 | 0.933 / 0.932 | 37.95 / 37.90 | 0.968 / 0.968 | 39.30 / 39.29 | 0.974 / 0.974 |

# B  Additional Experiments for Feat-Bias

In this section, we present additional experiments to validate the effectiveness of Feat-Bias (Sec.3). Following prior work[62, 19, 15], we evaluate our method on INR classification tasks using MNIST and F-MNIST datasets. In addition to the baselines discussed in Sec.D, we include comparisons with recent methods including Inr2Vec[28], DWS [33], and NG-GNN [23]. Feat-Bias$_{[ViT]}$ and Feat-Bias$_{[DINOV2]}$ represent our method using features extracted from ViT [10] and DINOv2 [34], respectively. We maintain identical configurations as described in Sec. D, employing a $3 \times 256$ MLP downstream network, training for 1000 iterations with a cosine scheduler and learning rate of $1e-3$. The quantitative results are presented in Tab. 6. Our Feat-Bias$_{[DINOV2]}$ achieves the highest classification accuracy on both datasets, surpassing the recent state-of-the-art MWT-L. Notably, on F-MNIST, our method improves accuracy from 87.32% to 92.38%, demonstrating significant advancement in INR downstream tasks.

Table 6: INR classification on MNIST and Fashion-MNIST Datasets

| Method | MNIST | F-MNIST | Method | MNIST | F-MNIST |
|---|---|---|---|---|---|
| MLP | 17.55±0.01 | 19.91±0.47 | Inr2Vec [28] | 23.69±0.10 | 22.33±0.41 |
| NFN$_{NP}$ [62] | 78.50±0.23 | 68.19±0.28 | NFN$_{HNP}$ [62] | 79.11±0.84 | 68.94±0.64 |
| DWS [33] | 85.71±0.57 | 67.06±0.29 | NG-GNN [23] | 91.40±0.60 | 68.00±0.20 |
| ScaleGMN [19] | 96.57±0.10 | 80.46±0.32 | ScaleGMN-B [19] | 96.59±0.24 | 80.78±0.16 |
| WT [15] | 93.08±2.26 | 73.81±1.43 | MWT$_{Mid\text{-}Task}$ [15] | 95.57±0.30 | 77.23±0.56 |
| MWT [15] | 96.58±0.32 | 83.86±0.91 | MWT-L [15] | 98.33±0.13 | 87.32±0.16 |
| Feat-Bias$_{[ViT]}$ | 95.79±0.04 | 90.13±0.49 | Feat-Bias$_{[DINOV2]}$ | 98.48±0.05 | 92.38±0.06 |

# C  Additional Experiments for Ablation Studies

We present additional ablation studies to further verify the source of *spatial aliasing*. All experimental settings strictly align with those in Sec. 4.3. Tab 7 present ablation result for spatial aliasing on Kodak dataset, demonstrating consistency with the findings presented in Tab. 3.

Table 7: Ablation study for *spatial aliasing* (Kodak Dataset)

| Settings | standard | w/o coord. sym. | w/o act. sym. $(+1)$ | w/o act. sym. $(-1)$ |
|---|---|---|---|---|
| Full-Bias | 35.23 / 0.929 / 0.089 | 32.23 / 0.880 / 0.191 | 30.38 / 0.838 / 0.257 | 30.44 / 0.840 / 0.254 |
| Bias-Free | **15.10 / 0.626 / 0.461** | 32.17 / 0.877 / 0.200 | 30.11 / 0.832 / 0.258 | 30.11 / 0.832 / 0.259 |

# D  Additional Experiments for classification task

**Experimental Settings.** In this section, we further evaluate the effectiveness of our proposed Feat-Bias (Sec. 3) on INR downstream tasks. The downstream network of Feat-Bias consists of a lightweight $3 \times 256$ MLP classifier. Since the CIFAR-100 dataset is more complex than CIFAR-10, the number of iterations is changed to 5000, using a cosine scheduler with a learning rate of $1e-3$. In addition, the iteration count of MWT has also been modified to 20 epochs. In tab 8, we list the performance metrics at 10 epochs (i.e., the original default setting) and 20 epochs. All experiments are repeated three times, reporting both mean and standard deviation.

**Quantitative Results.** Tab. 8 shows the classification results for CIFAR-100 dataset [24], where we also adopt Implicit-Zoo's default configuration with a $1 \times 64$ SIREN network like the settings in the main text to ensure consistent experimental conditions across all comparisons. Since Implicitly Zoo does not provide the INR dataset of CIFAR-100, only MWT is compared with our method here. As evidenced by the results, our method achieves superior performance across accuracy, precision, and F1 score metrics meanwhile greatly reduces the training time,further demonstrating Feat-Bias's effectiveness in INR post-processing tasks.

Table 8: INR classification on CIFAR-100 Datasets

| Method | Classification Task | | | | | INRs | |
| --- | --- | --- | --- | --- | --- | --- | --- |
| | Accuracy (%) ↑ | Precision (%) ↑ | F1 ↑ | Time ↓ | Params (kilo #) ↓ | PSNR (dB) ↑ | SSIM ↑ |
| WT$_{10\text{epochs}}$ [15] | $12.21_{\pm0.89}$ | $10.90_{\pm0.62}$ | $10.09_{\pm0.87}$ | $73.52_{\pm1.78}$ (min) | 261 | 32.33 | 0.942 |
| WT$_{20\text{epochs}}$ [15] | $15.83_{\pm0.71}$ | $13.82_{\pm0.85}$ | $13.84_{\pm0.79}$ | $151.78_{\pm2.93}$ (min) | 261 | 34.89 | 0.964 |
| MWT$_{\text{Mid-Task-10epochs}}$[‡] [15] | $15.81_{\pm0.44}$ | $14.94_{\pm0.86}$ | $13.86_{\pm0.74}$ | $83.41_{\pm1.55}$ (min) | 261 | 30.17 | 0.910 |
| MWT$_{\text{Mid-Task-20epochs}}$[‡] [15] | $19.87_{\pm0.21}$ | $17.88_{\pm0.11}$ | $18.06_{\pm0.15}$ | $169.42_{\pm2.80}$ (min) | 261 | 33.06 | 0.948 |
| MWT$_{10\text{epochs}}$ [15] | $19.46_{\pm0.88}$ | $18.09_{\pm0.58}$ | $17.67_{\pm0.69}$ | $83.74_{\pm0.97}$ (min) | 261 | 24.45 | 0.752 |
| MWT$_{20\text{epochs}}$ [15] | $23.29_{\pm0.57}$ | $21.56_{\pm0.67}$ | $21.78_{\pm0.65}$ | $168.0_{\pm1.27}$ (min) | 261 | 25.80 | 0.806 |
| Feat-Bias$_{[\text{ViT}]}$ | $74.83_{\pm0.07}$ | $75.03_{\pm0.09}$ | $74.79_{\pm0.07}$ | $144.70_{\pm1.83}$ (sec) | 151 | / | / |
| Feat-Bias$_{[\text{DINOV2}]}$ | $78.51_{\pm0.16}$ | $78.71_{\pm0.13}$ | $78.46_{\pm0.15}$ | $145.38_{\pm1.65}$ (sec) | 151 | / | / |

‡ denotes $\omega_{task}$ reported in MWT [15].

# E  Additional visualizations for *spatial aliasing*

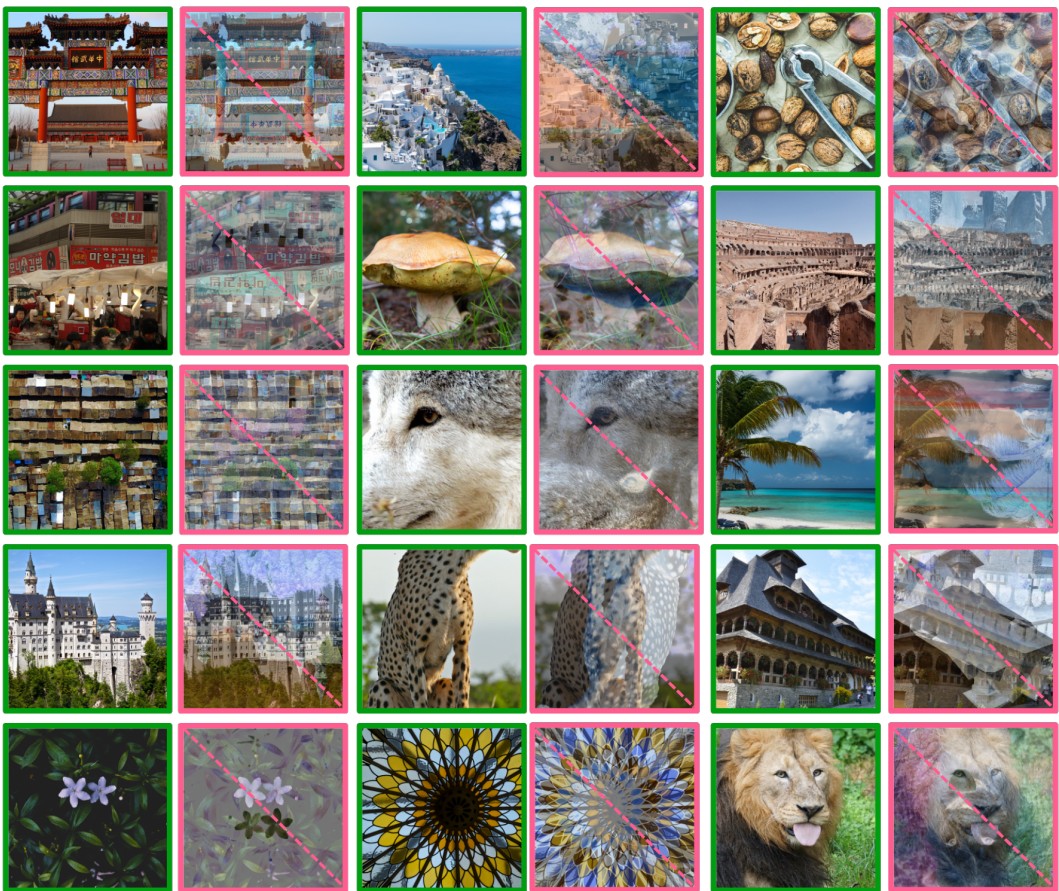

Figure 5: Visualization for *spatial aliasing*: reconstructed signals demonstrate central symmetry, manifesting as aliasing artifacts between distinct regions of the input. Green and Pink denote the ground truth and reconstructed signals with *spatial aliasing*, respectively.

