# OpenReview forum: "Understanding Bias Terms in Neural Representations"
_NeurIPS.cc/2025/Conference — NeurIPS 2025 poster_

### Official Review · Reviewer_aC5b · 2025-06-13

**Clarity:** 3
**Significance:** 3
**Originality:** 2
**Rating:** 4
**Confidence:** 4

**Summary:**

This paper studies the effect of bias term in commonly used INRs which utilize periodic activation functions. In contrast to piece-wise linear activation functions like ReLU, the bias term does not have the same influence on the network behavior and prediction performance as for commonly used ReLU-networks. The authors find that while the bias term has less influence on overall performance, completely omitting it leads to "spatial aliasing" an infolding of spatial content. As a solution, they propose using a frozen bias term in the first layer, which is distilled from a larger ViT. This approach aims to shift the INR's symmetry, thereby eliminating the infolding spatial content.

**Questions:**

1. Could the authors add a comparison between Feat-Bias and a randomly sampled bias vector which follows different distributions (e.g., uniform-like, Gaussian-like)? Is the use of a feature vector from a ViT necessary?
2. While the authors provide several examples for "spatial aliasing" for bias free IRNs, it would be interesting to visualize the reconstruction results yielded by networks with frozen biases like done with Feat-Bias. For a fair comparison it would be even more interesting to qualitatively compare standards IRN with periodic activation function with learned biases, fixed biases and Feat-Bias.
3. The visualization of the spatial aliasing suggested that the diagonal line is the center line, however, it seems as the artifacts are negatives of the original one flipped upside down, is there a reason why the authors used a diagonal line?
4. In the context of INRs, [a,b] used INRs, or more precisely, Hypernetworks, to learn CNN kernel weights. Would the authors think that this kind of inherent symmetry might even be beneficial for kernel learning with INRs?

[a] David W. Romero, Robert-Jan Bruintjes, Jakub Mikolaj Tomczak, Erik J Bekkers, Mark Hoogendoorn, and Jan van Gemert. Flexconv: Continuous kernel convolutions with differentiable kernel sizes. In International Conference on Learning Representations, 2022a. URL https://openreview.net/forum? id=3jooF27-0Wy.

[b] Grabinski, Julia, Janis Keuper, and Margret Keuper. "As large as it gets-studying infinitely large convolutions via neural implicit frequency filters." Transactions on Machine Learning Research 2024 (2024): 1-42.

**Ethical Concerns:**

["NO or VERY MINOR ethics concerns only"]

**Final Justification:**

Based on the clarifications provided by the authors, I have decided to raise my score.

**Limitations:**

As the authors state, a theoretical explanation and proof are missing. Therefore, the evidence relies solely on the empirical study and the intuition regarding the inherent symmetry induced by coordinates and the periodic activation function. However, the qualitative empirical analysis and evidence are incomplete and should be added to compensate for the missing theoretical evaluation.

Since the authors don't present qualitative results for their approach, it's unclear whether the spatial aliasing problem is fully resolved or merely shifted. This is particularly relevant for the Feat-Bias configuration (a frozen bias term only in the first layer), which could potentially lead to shifted artifacts because the bias term isn't adapted during learning.

If the authors can thoroughly evaluate their configuration and demonstrate through qualitative empirical evaluation (going beyond the current performance evaluation) that adapting the bias term to a fixed Feat-Bias leads to fewer artifacts, and thus should become the new standard in INRs, I would be happy to raise my score.

**Quality:**

2

**Strengths And Weaknesses:**

Strengths:
- Novelty: The study of the bias term in INRs utilizing periodic activation functions, as opposed to piece-wise linear ones, is both novel and compelling.
- Efficiency: Employing a fixed bias term in the first layer and eliminating additional bias terms in subsequent layers can lead to savings in computational resources and memory consumption.
- Clarity: The paper is well-written, with a clear and easy-to-follow line of argumentation.

Weaknesses:
1. While the intuition regarding spatial aliasing is compelling, the paper lacks a theoretical analysis to substantiate these insights.
2. Although the authors provide good empirical evidence for spatial aliasing when a bias term is omitted, they do not offer qualitative empirical evidence demonstrating that their proposed approach (using a fixed bias term only in the first layer) effectively eliminates spatial aliasing. Thus, besides the empirical performance evaluation, qualitive results should be provided.
3. The usage of a compressed feature vector from a larger ViT, while conceptually interesting, raises questions about its necessity and whether the feature vector genuinely remains useful given the potential for inherent aliasing introduced by the compression pipeline of this feature vector.
4. The authors claim that the bias term in INRs with periodic sine activation results in a phase shift (line 111). However, this potentially interesting property is not further discussed, despite its possible role in the observed spatial aliasing.

---

> ### Author Rebuttal · Authors · 2025-07-30
>
> We sincerely thank the reviewer for your valuable comments and suggestions. In addressing your insightful comments, we provide the following clarifications:
>
> - **Lack of Formal Mathematical Proof.** We admit that our work lacks rigorous mathematical proof to explain our insights. In this paper, we challenge the conventional practice of implementing full-bias INR architecture in this field and explain the role of bias terms through intuitive analysis and empirical evidence, proposing that frozen first-layer bias terms should be the new paradigm with efficient parameters and improved reconstruction quality. Many foundational papers in the INR field are combinations of intuitive analysis, observation, and effective implementation without mathematical proof (e.g., SIREN [1], FINER [2], WIRE [3], Trans-INR [4]). Their theoretical frameworks might be later formulated in separate works published 2-3 years after (e.g., Sampling [5], Algebra [6] ). Our current objective is to share our discoveries with the research community, aiming to inspire further exploration in both theoretical frameworks and practical advancements in this domain. We intend to incorporate formal mathematical proof in our future work.
> - **Qualitative Results for the Elimination of Spatial Aliasing.** Due to the constraints of the rebuttal page, we are unable to provide additional visually represented qualitative results to illustrate the benefits of the frozen first-layer bias terms mechanism. However, we have included an illustrative visual representation of the impact of frozen bias terms in INRs in Fig. 1 (line 39) in the paper. From this, we can infer that, with the exception of the bias-free configuration (#6) and the sole implementation of bias in the output layer (#5), other bias configurations exhibit the elimination of spatial aliasing. Our proposed approach of solely implementing input bias (#2) achieves optimal reconstruction performance, outperforming the conventional full-bias architecture (#1). In the final version, we plan to incorporate additional visual comparisons and enhance the analysis of local image details.
> - **Concerns Regarding Phase Shift.** In line 111 of the manuscript, we highlighted that bias terms in sine activation primarily introduce phase shifts rather than enforcing strict gating states, suggesting that the influence of bias terms in periodic activation functions should manifest significantly smaller effects compared to traditional ReLU-based activation functions. While we recognize that such phase shifts may have unique implications on the final reconstruction, we have not identified a clear and explicable relationship between the shifted phase and the ultimate reconstructed outcome, even after applying recent frameworks designed to explain the properties of INRs [7]. We value this insight greatly and intend to delve deeper into this aspect in our future work.
> - **Comparing Feat-Bias with Other Sampling Methods.** The results of this experiment have been detailed in the response to **Reviewer x6DW**, presented under the bolded title **Additional Experiments for Feature Sampling.**
> - **Is the use of a feature vector from a ViT necessary?** For INR post-processing task, such as INR classification, the utilization of a feature vector is essential. In the specified lines 202 to 220 (Sec. 3), we posit that Feat-bias can significantly enhance INR classification performance by leveraging fixed bias terms as optimal repositories for high-level features derived from established pre-trained encoders. These bias terms function as a distillation mechanism, transferring high-level features from extensive pre-trained networks to more compact input-layer bias spaces. This process allows each INR to capitalize on the representation capabilities of a large pre-trained network, resulting in substantial advancements when integrating additional networks for downstream INR tasks. Regarding INR reconstruction, the distillation of the feature vector is deemed unnecessary. The pivotal factor lies in the configuration of bias term **presence** (bias terms exclusively in input layers v.s. bias terms across all layers), rather than considerations of adaptiveness or inclusion of high-level feature information, as these aspects do not significantly impact the reconstruction ability, demonstrating near-identical performance.
> - **Presentation Regarding Diagonal Line.** We employ diagonal lines to showcase the impact of spatial aliasing, emphasizing that this aliasing stems from center symmetry rather than horizontal (x-axis) or vertical (y-axis) symmetry. Additionally, we aim to illustrate that the artifacts are essentially inverted negatives of the original one flipped both vertically and horizontally.
> - **Potential Benefits for INR-based Kernel Learning.** The insights derived from our analysis of bias have the potential to enhance INR-based kernel learning under the following conditions: (#1) symmetry in input structure and (#2) symmetry in activation functions (such as SIREN and FINER). While the original configurations of FlexConv (ICLR 2022) and NIFF (TMLR 2024) met condition #1 but did not fulfill condition #2 due to FlexConv utilizing MFN as the INR backbone and NIFF employing a PE+ReLU network, neither of which are periodically activated INRs. Given that FINER [2] has exhibited superior expressive capabilities compared to MFN and PE+ReLU, integrating FINER or SIREN into FlexConv and NIFF stands to benefit from the insights presented in our study.
> - **Comprehensive Evaluation of the Proposed Bias Mechanism**. Here, we present additional experimental results demonstrating that exclusively applying frozen bias terms to input layers consistently outperforms fully biased networks. This approach is poised to set **new standards in the INR community**, promising enhanced efficiency in storage and reconstruction capabilities.
>     - **Supplementary Material Inclusion.** We offer further validation of our empirical investigations (Section 2.2) and the proposed bias mechanism through additional datasets and bias configurations, including more comprehensive analyses of bias terms and the impacts of gradient optimization. Detailed settings and results are accessible in Section A of the supplementary material.
>     - **Additional Evaluation on Other Modalities**
>         - **1D Audio Fitting Task**. Fitting 1D audio data can be formulated as $F_\theta(\mathbf{x}): (t) \mapsto (a)$, where $a$ represents the amplitude value at time step $t$.
>         For this task, we follow the settings of [8], utilizing LibriSpeech dataset and configuring both $\omega$ and $\omega_0$ to 100 in the SIREN architecture.
>
>
>             |  | SI-SNR ↑ | STOI ↑ | MSE (e-4) ↓ |
>             | --- | --- | --- | --- |
>             | Full-bias  | 11.21 | 0.896  | 2.835  |
>             | Input Bias (Ours) | 11.72 | 0.904 | 2.239 |
>         - **2D Text Fitting Task**. Fitting 2D synthesized text image data can be formulated as $F_\theta(\mathbf{x}): (x,y) \mapsto (r,g,b)$. The dataset in this experiment was obtained from PE. We maintain all settings consistent with settings described in Sec.4.1 (line 223 -230).
>
>
>             |  | PSNR ↑ | SSIM ↑ | MSE (e-3) ↓ |
>             | --- | --- | --- | --- |
>             | Full-bias [T=1k] | 33.94 | 0.983 | 1.964 |
>             | Input Bias (Ours) [T=1k] | 34.38 | 0.985 | 1.775 |
>             | Full-bias  [T=3k] | 40.11 | 0.994 | 0.490 |
>             | Input Bias (Ours) [T=3k] | 40.49 | 0.996 | 0.443 |
>         - **3D Video Fitting Task**. Fitting 3D video data can be formulated as $F_{\theta}: (x,y,t) \mapsto (r,g,b)$.  Following the setting of DINER [9], the experiments were evaluated on the video from the UVG dataset.  Each scenario was trained for 100 epochs.
>
>
>             |  | PSNR ↑  | SSIM ↑  |
>             | --- | --- | --- |
>             | Full-bias  | 26.97 | 0.981 |
>             | Input Bias (Ours)  | 27.35 | 0.984 |
>         - **3D Shape Fitting Task**. We used Signed Distance Fields to represent 3D shapes. The fitting task can be formulated as $F_\theta(\mathbf{x}): (x,y,z) \mapsto (s)$, where $(x,y,z)$ represents the coordinate of given points and $s$ denotes the signed distance to the surface. Following [10], we employed an $8\times256$ MLP with SIREN architecture. We evaluated our method on the Stanford 3D Scanning Repository. The total iterations were set to $T=5,000$. The results are are as follows:
>
>
>             |  | IoU ↑ | CHD ↓ |
>             | --- | --- | --- |
>             | Full-bias  | 0.949  | 1.45e-6  |
>             | Input Bias (Ours)  | 0.954 | 1.41e-6 |
>         - **Conclusion**. The above quantitative results evaluated across different modalities demonstrate the enhanced reconstruction achieved by implementing frozen bias terms solely in the input layer (Input Bias), further supporting the main findings of our paper and challenging the conventional practice of implementing full-bias INR architecture in the community.
>
> [1] Implicit neural representations with periodic activation functions. In NeurIPS, 2020.
>
> [2] FINER: flexible spectral-bias tuning in implicit neural representation by variable-periodic activation functions. CVPR, 2024.
>
> [3] WIRE: wavelet implicit neural representations. CVPR, 2023
>
> [4] Learning Transferable Features for Implicit Neural Representations. In NeurIPS, 2024.
>
> [5] A Sampling Theory Perspective on Activations for Implicit Neural Representations. ICML, 2024.
>
> [6] Implicit Neural Representations and the Algebra of Complex Wavelets. ICLR 2024.
>
> [7] Explaining the Implicit Neural Canvas: Connecting Pixels to Neurons by Tracing their Contributions. CVPR 2024
>
> [8] EVOS: Efficient Implicit Neural Training via EVOlutionary Selector. CVPR, 2025.
>
> [9] DINER: Disorder-Invariant Implicit Neural Representation. CVPR, 2024.
>
> [10] Nonparametric Teaching of Implicit Neural Representations. ICML, 2024.

---

> > ### Comment · Reviewer_aC5b · 2025-08-04
> >
> > I appreciate the authors' efforts in their rebuttal to address my concerns. However, my original skepticism remains largely unchanged.
> >
> > My core concern regarding the selection of fixed ViT feature vectors is that the compression pipeline inherently introduces aliasing artifacts, which compromises the integrity of the feature vector. The authors' claim of its superiority is not convincingly supported, especially when a similar outcome could potentially be achieved with a feature vector that doesn't require compression.
> >
> > Additionally, the response to my question about phase shifts was insufficient. The authors merely state that their approach has less of an effect compared to standard gating mechanisms, but they fail to address the potential for issues caused by phase shifts that might occur under different scales or spatial shifts.
> >
> > Finally, I find the inclusion of additional, unsolicited experimental results in the rebuttal to be inappropriate. Authors should respond to reviewer feedback based on the submitted work, not introduce new data that was not ready during the initial submission. This practice is concerning and does not contribute to a fair review process.
> >
> > In summary, my original concerns have not been resolved, and therefore, my score will remain unchanged.

---

> > > ### Author Response · Authors · 2025-08-04
> > >
> > > - We sincerely appreciate your insightful evaluation of our responses and the valuable feedback. In our reply, we aim to offer further elucidation on the raised concerns.
> > > - **(1) Compression of Fixed ViT Feature Vectors**
> > >     - Our approach (Feat-bias) involves transferring high-level feature vectors into the bias terms space within the input layer, presenting a direct and efficient solution for INR classification tasks. This strategy is inspired by our Observations #1 and #2, where the stability of bias terms, unaffected by backpropagation, allows them to retain extracted high-level features from pre-trained encoders. By harnessing the expressiveness of these features, we achieve notable performance improvements in INR **Post-processing task**.
> > >     - Importantly, the incorporation of feature vector compression does **NOT** compromise **Reconstruction** performance. Initializing the first-layer bias with methods such as $x \sim U(-\frac{1}{\sqrt{n}}, \frac{1}{\sqrt{n}})$, $x \sim \text{Gauss}(-\frac{1}{\sqrt{n}}, \frac{1}{\sqrt{n}})$, or $x = \text{Scale}(\text{Downsample}(\text{ViT}(\mathbf{y} _ \text{G.T.})), \frac{1}{\sqrt{n}}, \frac{1}{\sqrt{n}})$ (compressing features to a specific dimension → scaling) yields nearly identical reconstruction performance (within an average of ±0.01 dB) with or without bias term backpropagation. This is attributed to the effectiveness of only the **presence** of first-layer bias can achieve the best performance compared with full-bias mechanism. Our Feat-bias mechanism significantly enhances INR **post-processing** performance without compromising **reconstruction** (in comparison to the mechanism of only fixed first-layer bias but without featuring infusion), surpassing traditional full-bias INR architectures in reconstruction.
> > >     - In the context of **reconstruction** tasks, the key factor for enhancing performance and alleviating spatial aliasing effects lies in achieving symmetry around 0 with the fixed first-layer bias construction method (be it Gaussian distribution, uniform distribution, ours: downsampling from ViT followed by rescaling, etc.) Further details on this can be found in the *Response to Reviewer sknb*. The utilization of compressed fixed feature vectors aims to concurrently improve performance in both INR *post-processing* and *reconstruction*, while accommodating the need for flexible neuron numbers in the initial layer.
> > >     - In conclusion, we respectfully believe that your statement regarding "*The authors' claim of its superiority is not convincingly supported, especially when..*." may contain some misunderstandings. If you have any additional questions regarding this matter, please feel free to raise them at any time.

---

> > > > ### Author Response · Authors · 2025-08-04
> > > >
> > > > - **(2) Phase Shift**
> > > >     - **Reasons for Mentioning Phase Shift.** The mention of phase shift in line 111 of the manuscript is intended to offer supplementary insight into the comparatively minor impact of bias terms in periodic activation functions, in contrast to traditional ReLU-based activations. This discussion serves as an additional *prologue* to examining the role of bias terms in INRs with robust motivations and a cohesive narrative. It is important to note that this reference serves as a subtle introduction in our Intuitive Analysis (Sec. 2.1) and does not represent the central focus of our paper.
> > > >     - **Potential Issues Caused by Phase Shifts.** As stated in our rebuttal, we have indeed **NOT** identified a clear and explicable relationship between shifted phases and the fitting dynamics of INRs. This lack of correlation is evident from our empirical investigations (Sec. 2.2) and Additional Experiments (Fig.2 (2) and Sec.A in the supplementary material), where **(Observation #2) The gradient optimization of bias has minimal impact on INRs.** Specifically, the output value of the $l$-th layer is represented as $\mathbf{z} _ l= f_l(\mathbf{z} _ {l-1}) =  \sin(\omega_0\alpha^l(\mathbf{W} _ l \mathbf{z} _ {l-1}+\mathbf{b} _ l)$, where $\mathbf{W} _ l$ and $\mathbf{b} _ l$ denote the weights and bias at layer $l$. The behavior of $\mathbf{W} _ l$ and $\mathbf{b} _ l$ during INR fitting can be comprehended in terms of **Frequency** and **Phase Shifts**. However, based on our empirical findings, the dynamics of bias terms have minimal impact on INRs, indicating that phase shifts have little influence on INRs, at least in the general training of INRs. In other words, what truly mitigates spatial aliasing is the **initialization of a fixed phase rather than the dynamic shifting of the phase**.
> > > >     - We highly appreciate the reviewer's perceptive notion that such phase shifts should have distinctive effects. However, our rigorous experiments indicate otherwise. We speculate that such influence may manifest under additional regularization of INR architecture, which we consider an exciting direction for future investigation.
> > > > - (3) **Inclusion of Additional Experimental Results**
> > > >     - We would like to clarify that our **Comprehensive Evaluation of the Proposed Bias Mechanism** in our rebuttal is in response to the need for a more comprehensive assessment of the advantages of our method (as outlined in the final part of the Limitation section in the original review, specifically in the paragraph *If the authors can thoroughly evaluate…*). Due to the constraints of the rebuttal policy, we are unable to provide external links to visually showcase qualitative results that highlight the benefits of our method. Therefore, we are considering demonstrating our strengths and potentials through the (1) inclusion of supplementary material and (2) evaluations across various modalities. This includes the ability to generalize to different modalities as a crucial aspect of qualitative empirical evaluation and further configuration testing. We sincerely assure you that our original intention was solely to address your request and not to compromise the fairness of the review process.
> > > >
> > > > We really appreciate your constructive feedback to our submission and your continued engagement during the discussion period! If there are any remaining issues, we would be happy to provide additional clarifications.
> > > >
> > > > Thank you very much for your time and consideration!

---

> > > > ### Comment · Reviewer_aC5b · 2025-08-04
> > > >
> > > > Thank you for your response.
> > > >
> > > > The core of my concern remains a fundamental ambiguity in the presented results. The authors state in lines 210ff that the extracted ViT feature vectors "approximately follow a uniform distribution." This raises a critical question: is the observed benefit derived from the specific high-level features embedded within the ViT vector, or is it simply a consequence of using a stabilizing bias term with an approximately uniform distribution?
> > > >
> > > > To convincingly demonstrate that the ViT features are indeed more useful than a generic stabilizing factor, I would find it compelling to see an experiment where the ViT features are replaced with a synthetically generated feature vector that also follows an approximate uniform distribution.
> > > >
> > > > Without such a comparison, it is not clear whether the positive effect is due to the ViT's semantic content or merely its statistical properties, which stabilize the training process.

---

> ### Author Response · Authors · 2025-08-04
>
> We genuinely appreciate your insightful feedback and active participation during the discussion period.
>
> We indeed highlighted that “*we find that feature vectors extracted from pre-trained encoders approximately follow a uniform distribution*” (line 210). This observation underscores the importance of the feature-based initialization for the first-layer bias terms ($x = \text{Scale}(\text{Downsample}(\text{ViT}(\mathbf{y} _ \text{G.T.})), \frac{1}{\sqrt{n}}, \frac{1}{\sqrt{n}})$), as it allows us to “*maintain reconstruction performance while significantly enhancing INRs’ feature representation capabilities*” (line 213).
>
> **Response to the Question.**
>
> > *Is the observed benefit derived from the specific high-level features embedded within the ViT vector, or is it simply a consequence of using a stabilizing bias term with an approximately uniform distribution?*
> >
>
> As mentioned in our rebuttal, the observed **classification** (post-processing) benefits stem from high-level ViT features, while the **reconstruction** benefits result from the mechanism of utilizing frozen bias terms exclusively in the input layer with an approximately uniform distribution. Since the extracted features naturally exhibit an approximate uniform distribution, rescaling them with $(-\frac{1}{\sqrt{n}}, \frac{1}{\sqrt{n}})$ maintains the reconstruction enhancement from fixed input-layer bias terms, leading to simultaneous improvements in both INR reconstruction and post-processing classification performance. We have summarized the experimental results as follows:
>
> | Task | (1) $\mathbf{b} _ 1  = \text{Scale}(\text{Down}(\text{ViT}(\mathbf{y} _ \text{G.T.})), \frac{1}{\sqrt{n}}, \frac{1}{\sqrt{n}}))$ | (2)  $\mathbf{b}\_1 \sim U(-\frac{1}{\sqrt{n}}, \frac{1}{\sqrt{n}})$ | (3) Conventional Full-bias Arch. | Corresponding Comparison  |
> | --- | --- | --- | --- | --- |
> | Reconstruction | 36.63 / 0.962 / 0.019 | 36.64 / 0.963 / 0.019 | 36.19 /  0.959 / 0.023 | / |
> | Classification | 93.78 / 93.80 / 93.78 | 11.21 / 11.19 / 11.20 | 11.20 / 11.18 / 11.17 | MWT [1]: 46.94 / 46.48 / 46.51;
> ZOO [2]:  81.57 / 81.53 / 81.51 |
>
> **Experimental Settings**. Both experiments #1 and #2 implement proposed mechanism of utilizing frozen bias terms exclusively in the input layer. Experiment #1 corresponds to Feat-bias (Sec.3, line 190), while experiment #2 represents the well-adopted bias initialization method from [3]. Experiment #3 refers to the conventional Full-bias architecture widely adopted within the INR community. The experimental configurations for reconstruction and classification align with Sec. 4.1 and Sec. 4.2. The reported metrics for the reconstruction task include PSNR (↑) / SSIM (↑) / LPIPS (↓), while for the classification task, the metrics consist of Accuracy (↑) / Precision (↑) / F1 (↑).
>
> **Analysis of Results**. As illustrated in the table above, our Feat-bias (#1) demonstrates the ability to enhance performance in both INR reconstruction (in comparison to full-bias) and post-processing classification (when compared to complex state-of-the-art frameworks such as MWT[1] and ZOO[2]). While the mechanism of #2 can achieve an improvement in reconstruction quality with more efficient memory usage, it faces challenges related to the interpretability of neural parameters for downstream processes, leading to subpar performance in INR classification.  In contrast, Feat-bias can effectively utilize this stabilized bias space to represent high-level features extracted from pre-trained encoders, thereby significantly enhancing the performance of INR post-processing without compromising the reconstruction benefits identified in our findings (Sec. 2.1, Sec. 2.2, Sec. 2.3).

---

> ### Author Response · Authors · 2025-08-04
>
> *(Following from the last page)*
>
> **Why does the inclusion of featured bias terms significantly enhance the performance of INR post-processing?** As explained in lines 202 to 220 (Sec. 3), we contend that Feat-bias can notably enhance INR classification performance by utilizing fixed bias terms as optimal storage units for high-level features extracted from well-established pre-trained encoders. These featured bias terms function as a form of distillation mechanism, transferring high-level features from large pre-trained networks to more compact input-layer bias spaces. Consequently, each INR model can benefit from the representational capacities of extensive pre-trained networks, leading to substantial improvements when integrating additional networks for downstream INR tasks.
>
> **Why does the inclusion of featured bias terms NOT provide additional enhancement for INR reconstruction?** It is important to reiterate that the improved reconstruction performance is attributed to the mechanism of utilizing frozen first-layer bias terms with a uniform distribution rather than the integration of high-level ViT features. This can be comprehended from the following perspectives:
>
> 1. The **high-level** features extracted from pre-trained encoders (ViT) exhibit limited influence on the **low-level** task of INR reconstruction. Additionally, the pre-trained network operates on **explicit** signal forms (such as images as input), which are not aligned with the **implicit** signal forms (such as coordinates as input) central to INR.
> 2. The **permutation symmetries** present in neural parameters impede the effectiveness of high-level features for INR reconstruction. In neural networks, permutation symmetries suggest that the rearrangement of neural parameters does not alter the overall behavior of the model. Bias terms, as integral components of neurons, also adhere to this property, wherein any rearrangement of bias connections to a specific neuron in the next layer does not impact the output. When high-level features are embedded into the bias terms of the first layer ($\mathbf{b} _ 1  = \text{Scale}(\text{Down}(\text{ViT}(\mathbf{y} _ \text{G.T.})), \frac{1}{\sqrt{n}}, \frac{1}{\sqrt{n}}))$), if we anticipate a positive effect on model fitting, the random permutation of these bias terms should yield a similar improvement in accordance with the permutation symmetries of the neural network. However, random permutation of ViT features would essentially eliminate the expressiveness of the pre-trained network, resembling sampling from an approximate uniform distribution ($\mathbf{b} _ 1 \sim U(-\frac{1}{\sqrt{n}}, \frac{1}{\sqrt{n}})$), which aligns with the standard bias term initialization. Consequently, the anticipated beneficial impact of featuring bias initialization is not realized, leading to nearly identical reconstruction performance between featuring initialization (#1 in the above table) and standard initialization (#2 in the above table), consistent with our experimental findings in the above table.
>
> [1] End-to-end implicit neural representations for classification. CVPR, 2025.
>
> [2] Implicit zoo: A large-scale dataset of neural implicit functions for 2d images and 3d scenes. In NeurIPS, 2024.
>
> [3] Implicit neural representations with periodic activation functions. In NeurIPS, 2020.

---

> > ### Comment · Reviewer_aC5b · 2025-08-06
> >
> > Thank you for your clarification.
> >
> > A point of confusion remains regarding the setup of Experiment #2. Since it is based on the work in [3], could you please clarify whether this experiment utilizes the full WIRE network architecture, or if it only adopts the initialization scheme from [3] while keeping the network structure and all other experimental variables consistent with Experiment #1?

---

> > > ### Author Response · Authors · 2025-08-06
> > >
> > > Thank you for your participation in the discussion!
> > >
> > > Firstly, we would like to clarify that reference [3] (*Implicit neural representations with **periodic** activation functions. In NeurIPS, 2020*) mentioned in the previous section refers to the SIREN network, not the WIRE network. SIREN utilizes periodically activated functions (sine) to represent signals, which marks a pioneering contribution in the field of INR. Given its widespread popularity and superior performance [3][4], our study and experiments focus on these sine-based architectures, which serve as the default setting for all experiments detailed in the manuscript and our previous response.
> > >
> > > Regarding the experiments outlined above, #1 and #2 are **strictly conducted under the same configuration** (SIREN network & first-layer frozen bias terms & other settings), with the only variation being the initialization method for the first-layer bias terms (#1 utilizes scaled feature-based initialization, while #2 adopts the default initialization in SIREN widely adopted by the INR community).
> > >
> > > We believe that the results presented in the table above effectively support our clarification and additional explanation in our response. If there are any remaining confusions, we would be happy to provide additional clarifications. Thank you very much for your time and effort!
> > >
> > > [3] Implicit neural representations with periodic activation functions. In NeurIPS, 2020.
> > >
> > > [4] FINER: flexible spectral-bias tuning in implicit neural representation by variable-periodic activation functions. CVPR, 2024.

---

> > > > ### Comment · Reviewer_aC5b · 2025-08-06
> > > >
> > > > Thank you for the clarification. My confusion regarding the use of reference [3] has been fully resolved.
> > > >
> > > > Based on this productive discussion, I have decided to raise my score. I would encourage the authors to include more qualitative results in the final submission.
> > > >
> > > > Furthermore, for future work, it would be highly intriguing to formally demonstrate the benefit of using one fixed bias term while omitting all other bias terms. This would provide a strong theoretical foundation for the internal mechanisms of IRNs and enable the community to better understand these models, thereby increasing their transparency.

---

> > > > > ### Author Response · Authors · 2025-08-06
> > > > >
> > > > > Dear Reviewer aC5b,
> > > > >
> > > > > We deeply appreciate your insightful feedback. We are pleased to engage in a discussion to address your concerns. In the final version, we will incorporate all clarifications and additional insights. Furthermore, we will carefully consider your suggestions for future work.
> > > > >
> > > > > Thank you for acknowledging the contributions of our work and for your efforts to enhance our work!
> > > > >
> > > > > Best regards,
> > > > >
> > > > > The Authors

---

### Official Review · Reviewer_3n6q · 2025-06-22

**Clarity:** 2
**Significance:** 2
**Originality:** 3
**Rating:** 4
**Confidence:** 4

**Summary:**

This paper investigates the role of bias terms in Implicit Neural Representations (INRs) and challenges the common assumption that biases primarily enhance the network’s nonlinearity. Through extensive analysis, the authors demonstrate that biases mainly serve to eliminate spatial aliasing artifacts rather than increase model expressiveness. Based on this insight, the paper proposes a novel bias modulation technique called Feat-Bias, which strategically leverages bias terms to reduce aliasing effects in INR reconstructions.

Unfortunately, the paper is poorly written and difficult to follow. Although the authors emphasize reconstruction quality, the experimental setup for this task is weak. Conversely, the experiments primarily focus on the INR classification task. However, the section “3 Application: Featuring INRs with Bias Terms” is unclear and poorly organized. Furthermore, the key component, Feat-Bias, is insufficiently explained. In addition, Figure 4 is not referenced anywhere in the manuscript.

**Questions:**

1. The structure of the paper should be improved. The authors need to clearly specify which task is central to the paper—reconstruction or classification—and precisely describe the key architectural components.

2. A theoretical justification for the proposed approach should be included.

3. Additional experiments on diverse datasets and with different architectures are necessary.

4. The Feat-Bias method is unclear and insufficiently described.

5. Figures and visualizations should better emphasize the main advantages of the paper.

**Ethical Concerns:**

["NO or VERY MINOR ethics concerns only"]

**Final Justification:**

Na podstawie wyjaśnień i rozszerzonych wyników eksperymentalnych przedstawionych w odpowiedzi, podniosłem swoją ocenę. Zdecydowanie zaleca się jednak, aby ostateczna wersja artykułu zawierała następujące elementy. Po pierwsze, powinna zawierać krótką dyskusję na temat wpływu różnych funkcji aktywacji. Po drugie, bardziej szczegółowe i jednoznaczne wyjaśnienie metody Feat-Bias, najlepiej poparte pseudokodem lub schematyczną ilustracją, znacznie poprawiłoby czytelność. Po trzecie, rozwinięcie teoretycznej motywacji stojącej za obserwacjami dotyczącymi terminów stronniczości pomogłoby lepiej ugruntować wkład w teorię. Wreszcie, wizualizacje powinny zostać ulepszone, aby wyraźniej podkreślić zalety proponowanego podejścia. Te uzupełnienia znacznie poprawiłyby czytelność, kompletność i ogólny wpływ pracy.

**Limitations:**

The authors have not adequately addressed their work's limitations and potential negative societal impact. To improve, they should explicitly discuss the scope and boundaries of their method, such as the types of data and tasks for which their approach may be less effective.

**Paper Formatting Concerns:**

No major formatting issues were found. The paper generally follows the NeurIPS 2025 formatting guidelines. However, the authors should double-check figure citations and consistency in section headings to ensure full compliance.

**Quality:**

2

**Strengths And Weaknesses:**

Strengths:

1. The main topic is very interesting.

2. The hypothesis in the paper is well defined.

Weaknesses:

1. The paper lacks theoretical justification for the proposed hypothesis.

2. The notation of (Observation #1), (Observation #2), and (Observation #3) is unclear. The authors should precisely formulate their theses and provide theoretical support.

3. Reconstruction quality should be evaluated on different types of datasets, such as videos, 3D objects, and sounds—not only images.

4. The baselines used for reconstruction tasks are weak. The authors should include comparisons with:

- FreSh: Frequency Shifting for Accelerated Neural Representation Learning (ICLR 2025)
- SPDER: Semiperiodic Damping-Enabled Object Representation (ICLR 2024)

5. The authors should consider more architectures, such as Fourier Feature Networks and Positional Encoding.

6. More activation functions should be tested.

7. The Feat-Bias method is unclear and poorly described in the paper. Additionally, Figure 4 is not cited anywhere in the main text.

8. Training curves in Figure 2 are incomplete. It is unclear what happens when the model is trained for 10,000 iterations.

9. Figure 1 is not convincing since the differences in PSNR are small, and I do not observe any visible difference in the images.

10. Perhaps dividing the reconstruction and original image for comparison could help illustrate the differences more clearly.

---

> ### Author Rebuttal · Authors · 2025-07-30
>
> We sincerely thank the reviewer for your valuable comments and suggestions, and here is our response for your concerns:
>
> - **Lack of Formal Mathematical Proof**. As stated in Sec. 6 (line 338), we admit that our work lacks rigorous mathematical proof to explain our insights. However, we respectfully disagree that this undermines the contribution of our work. We challenge the conventional practice of implementing full-bias INR architecture in this field and explain the role of bias terms through intuitive analysis and empirical evidence, proposing that frozen first-layer bias terms should be the new paradigm with efficient parameters and improved reconstruction quality. Many foundational papers in the INR field are combinations of intuitive analysis, observation, and effective implementation without mathematical proof (e.g., SIREN, FINER, WIRE, Trans-INR [1]). Their theoretical frameworks might be later formulated in separate works published 2-3 years after (e.g., Sampling [2], Algebra [3]). At this stage, we aim to contribute our findings to the research community first, and then stimulate further studies in this direction in both theoretical frameworks and practical improvements.
> - **Additional Evaluation on Other Modalities**
>     - **1D Audio Fitting Task**. Fitting 1D audio data can be formulated as $F_\theta(\mathbf{x}): (t) \mapsto (a)$, where $a$ represents the amplitude value at time step $t$.
>     For this task, we follow the settings of [4], utilizing LibriSpeech dataset and configuring both $\omega$ and $\omega_0$ to 100 in the SIREN architecture.
>
>
>         |  | SI-SNR ↑ | STOI ↑ | MSE (e-4) ↓ |
>         | --- | --- | --- | --- |
>         | Full-bias  | 11.21 | 0.896  | 2.835  |
>         | Input Bias (Ours) | 11.72 | 0.904 | 2.239 |
>     - **2D Text Fitting Task**. Fitting 2D synthesized text image data can be formulated as $F_\theta(\mathbf{x}): (x,y) \mapsto (r,g,b)$. The dataset in this experiment was obtained from PE. We maintain all settings consistent with settings described in Sec.4.1 (line 223 -230).
>
>
>         |  | PSNR ↑ | SSIM ↑ | MSE (e-3) ↓ |
>         | --- | --- | --- | --- |
>         | Full-bias [T=1k] | 33.94 | 0.983 | 1.964 |
>         | Input Bias (Ours) [T=1k] | 34.38 | 0.985 | 1.775 |
>         | Full-bias  [T=3k] | 40.11 | 0.994 | 0.490 |
>         | Input Bias (Ours) [T=3k] | 40.49 | 0.996 | 0.443 |
>     - **3D Video Fitting Task**. Fitting 3D video data can be formulated as $F_{\theta}: (x,y,t) \mapsto (r,g,b)$.  Following the setting of DINER, the experiments were evaluated on the video from the UVG dataset.  Each scenario was trained for 100 epochs.
>
>
>         |  | PSNR ↑  | SSIM ↑  |
>         | --- | --- | --- |
>         | Full-bias  | 26.97 | 0.981 |
>         | Input Bias (Ours)  | 27.35 | 0.984 |
>     - **3D Shape Fitting Task**. We used Signed Distance Fields to represent 3D shapes. The fitting task can be formulated as $F_\theta(\mathbf{x}): (x,y,z) \mapsto (s)$, where $(x,y,z)$ represents the coordinate of given points and $s$ denotes the signed distance to the surface. Following [5], we employed an $8\times256$ MLP with SIREN architecture. We evaluated our method on the Stanford 3D Scanning Repository. The total iterations were set to $T=5,000$. The results are are as follows:
>
>
>         |  | IoU ↑ | CHD ↓ |
>         | --- | --- | --- |
>         | Full-bias  | 0.949  | 1.45e-6  |
>         | Input Bias (Ours)  | 0.954 | 1.41e-6 |
>     - **Conclusion**. The above quantitative results evaluated across different modalities demonstrate the enhanced reconstruction achieved by implementing frozen bias terms solely in the input layer (Input Bias), further supporting the main findings of our paper and challenging the conventional practice of implementing full-bias INR architecture in the community.
> - **Central Task: Reconstruction or Classification?**
>     - Our paper does **NOT** aim to design a task-specific method. Instead, it **uncovers the overlooked role of bias terms in INRs** through analytical and empirical studies. Surprisingly, we discovered that bias terms minimally contribute to nonlinear representation, with their gradient propagation having little impact on performance. Based on these findings, we propose applying bias terms exclusively to input layers, challenging the conventional full-bias architecture. This adjustment consistently improves reconstruction performance, setting new standards in the INR community for more efficient storage and enhanced reconstruction.
>     - Our proposed Feat-bias for INR classification serves as a practical application of our understanding of bias terms in INRs. Since backpropagation of bias terms does not affect reconstruction, these values remain stable throughout the encoding process. This stability makes bias terms ideal for storing high-level features extracted from pre-trained encoders. Leveraging the expressiveness of these features, we achieve significant performance enhancements in post-processing tasks, offering a straightforward and effective solution in this field.
>     - **In summary**, improved reconstruction performance stems from eliminating redundant bias terms, while enhanced classification performance results from our optimized utilization of fixed bias terms, both rooted in our understanding of the role of bias terms in INRs.
> - **Application for Other Activation Functions.** Considering that sine-based periodic activation functions are widely used in the field of Implicit Neural Representations (INR) (SIREN being the most popular and FINER achieving the highest performance), we focus our study on INR architectures utilizing SIREN and its variations. Our analysis is also centered around the symmetry of these activations, a characteristic lacking in other activation functions (such as WIRE). It is observed that introducing a bias implementation in the first layer can lead to marginal improvements in models like PE and WIRE. However, the effectiveness of this enhancement is unstable, indicating the need for additional regularization in these architectures, a direction we plan to explore in future research. Given that FINER stands as the state-of-the-art architecture in INR reconstruction, we assert that the alignment of our work with FINER underscores the significance of our research endeavors.
> - **Absence of Comparison in INR Reconstruction.** Our method stands orthogonal to existing INR optimization methods, enabling seamless integration with these established frameworks without the need for direct comparisons. Over the past years, there have been approximately 40 papers focusing on improving the expressive capacity or fitting speed of INRs from diverse perspectives such as partition, meta-learning, transformation, and sampling. While we acknowledge the substantial challenge in evaluating the compatibility of our findings with all these diverse methods, we respectfully contend that the absence of comparisons with specific acceleration methodologies (e.g., FreSh) does **NOT** undermine the significance of our approach for the following reasons:
>     - The prevailing trend in INR studies is to benchmark against baselines within the same category (e.g., EVOS [4] versus INT [5] as both employ sampling-based acceleration techniques). Given that our method represents the first work in enhancing INRs by adjusting specific bias terms, the lack of existing baselines necessitates comparison.
>     - Indeed, FreSh (ICLR 2025) did not conduct comparative analyses across any of other INR optimization methods.
>     - Enhancing INRs' reconstruction quality is just one part of our contribution. Our main focus is highlighting the overlooked role of bias terms in INRs and questioning the prevalent full-bias architecture in the field.
>
>     We sincerely appreciate the suggestions provided and commit to incorporating experiments that align with various existing INR acceleration methods in the future work.
>
> - **Presentation Issues**
>     - **Fig. 2 are incomplete.** In actuality, **Fig. 2 is comprehensive**. The training dynamics under various bias configurations are examined over 5k iterations, following the methodology of previous studies [2][3][8]. The training curves indicate the convergence of the reconstruction process. Additionally, we conduct a re-implementation of this experiment with 10k iterations, showing a stable trend and reinforcing the conclusions drawn from the results illustrated in Fig. 2.
>     - **Visible Difference in Fig. 1.** We appreciate your valuable suggestions. To highlight the visible differences under various bias configurations, we will enhance the comparison of local details in the image in the final version.
>     - **Notation of Observations**. Given that these observations are not formalized within a mathematical framework, we designated them as Observation #1, #2, #3 for clarity and to facilitate a quicker comprehension of our empirical findings. These notations will be revised in the final manuscript.
>     - **Description of Feat-Bias Method**. Regarding the Feat-Bias method as a practical application of our interpretation of bias terms, we consider it a straightforward engineering solution. This method involves transferring encoded features to the bias terms in the input layer, hence, we have provided a concise treatment in the current version. A more detailed description of this method will be included in the appendix.
>
> [1] Learning Transferable Features for Implicit Neural Representations. In NeurIPS, 2024.
>
> [2] A Sampling Theory Perspective on Activations for Implicit Neural Representations. ICML, 2024.
>
> [3] Implicit Neural Representations and the Algebra of Complex Wavelets. ICLR 2024.
>
> [4] EVOS: Efficient Implicit Neural Training via EVOlutionary Selector. CVPR, 2025.
>
> [5] Nonparametric Teaching of Implicit Neural Representations. ICML, 2024.

---

> > ### Comment · Reviewer_3n6q · 2025-08-02
> >
> > Thank you for your detailed and thoughtful rebuttal. Based on your clarifications and the extended experimental results, I have decided to raise my overall score. However, I strongly recommend incorporating the following elements. First, please consider including a brief discussion on the impact of different activation functions. Second, a more detailed and explicit explanation of the Feat-Bias method, possibly supported by pseudocode or a schematic illustration, would significantly improve clarity. Third, expanding on the theoretical motivation behind your observations on bias terms would help ground the contribution more firmly in theory. Finally, I recommend enhancing the visualizations to highlight the advantages of your approach more clearly. These additions would significantly improve the clarity, completeness, and overall impact of the paper.

---

> ### Author Response · Authors · 2025-08-03
> **Response to Reviewer 3n6q**
>
> Dear Reviewer 3n6q,
>
> We appreciate your thoughtful review of our rebuttals. Your constructive feedback has enriched our study by incorporating more experimental validations and clearer presentation. In the final version, we will discuss additional activation functions, elaborate further on the theoretical motivation, and enhance clarity with pseudocode and additional illustrations.
>
> Thank you for recognizing the contributions of our work through the improved score. Your time and effort in assisting us to enhance our research are greatly appreciated.
>
> Best regards,
>
> The Authors

---

### Official Review · Reviewer_sknb · 2025-07-03

**Clarity:** 3
**Significance:** 3
**Originality:** 2
**Rating:** 4
**Confidence:** 5

**Summary:**

This paper investigates the infulence of the bias terms in INRs through both theoretical analysis and empirical validation. The authors demonstrate that bias terms in INRs primarily serve to eliminate spatial aliasing caused by symmetry from both coordinates and activation functions. To further address this issue, the paper introduces Feat-Bias, which initialize input-layer bias with high-level features extracted from pre-trained models.

Extensive experiments are performed on a representation task to validate the effectiveness of bias terms, and on a classification task using the CIFAR-10 dataset to demonstrate the performance of Feat-Bias.

**Questions:**

1. The classification experiment is conducted only on CIFAR-10, which may not sufficiently demonstrate the generalization ability of Feat-bias.

2. Could the authors offer a more formal mathematical characterization of how bias terms influence coordinate symmetries and contribute to the suppression of spatial aliasing?

3. Ablation experiment is limited.

**Ethical Concerns:**

["NO or VERY MINOR ethics concerns only"]

**Final Justification:**

The author followed our advice and conducted experiments on more datasets and provided additional mathematical formulas to illustrate his theory. Therefore, I decided to improve the final score.

**Limitations:**

yes

**Quality:**

2

**Strengths And Weaknesses:**

**Strengths**

1. Analyzing the role of bias terms in INRs from the perspective of spatial aliasing caused by symmetry is both novel and well-motivated.

2. Extensive experiments are carefully designed to validate the effect of bias terms.

**Weaknesses**

1. While the theoretical analysis is insightful, some of the claims would benefit from more formal proofs or clearer mathematical exposition.

2. The classification results are promising, but currently limited to CIFAR-10. Additional datasets would help assess the generality and robustness of Feat-bias in broader settings.

---

> ### Author Rebuttal · Authors · 2025-07-30
>
> We express our gratitude to the reviewer for the insightful feedback and suggestions provided. We are delighted that our work has been acknowledged as innovative, well-motivated, and rigorously tested. In response to your valuable comments, we offer the following clarifications:
>
> - **Additional Experiments for INR Classification**
>     - **Supplementary Material Inclusion**. Due to space constraints, we have included the INR classification results for CIFAR-10 in the main text. Moreover, we have conducted INR classification tasks on MNIST, F-MNIST, and CIFAR-100, with detailed outcomes presented in Sections B and C of the supplementary materials. Notably, our method surpasses the state-of-the-art MWT-L [1] in terms of classification accuracy across all datasets. Particularly noteworthy is the substantial enhancement in accuracy observed on F-MNIST, increasing from 87.32% to 92.38%. The summarized results for MNIST and F-MNIST are as follows:
>
>
>         |  | MNIST | F-MNIST |  | MNIST | F-MNIST |
>         | --- | --- | --- | --- | --- | --- |
>         | NFN | 78.50±0.23 | 68.19±0.28 | ScaleGMN-B | 96.59±0.24 | 80.78±0.16 |
>         | DWS | 85.71±0.57 | 67.06±0.29 | MWT  | 96.58±0.32 | 83.86±0.91 |
>         | NG-GNN | 91.40±0.60 | 68.00±0.20 | MWT-L | 98.33±0.13  | 87.32±0.16 |
>         | ScaleGMN | 96.57±0.10  | 80.46±0.32 | **Feat-bias (Ours)** | **98.48±0.05**  | **92.38±0.06** |
>     - **INR Classification on CIFAR-100 (Supplementary Material)**.The experimental results for CIFAR-100 are summarized below. Given the absence of code implementation for CIFAR-100 in Implicit-Zoo [2], our method is compared solely with MWT, showcasing superior performance in terms of accuracy, precision, F1 score metrics, while also significantly reducing training time. These outcomes further validate the effectiveness of Feat-Bias in INR post-processing tasks.
>
>
>         |  | Accuracy ↑ | Precision ↑ | F1 ↑ | Time ↓ | Parmas ↓ |
>         | --- | --- | --- | --- | --- | --- |
>         | WT | 15.83±0.71  | 13.82±0.85  | 13.84±0.79 | 151.78 ±2.93 (min)  | 261 |
>         | MWT [mid-task] | 19.87±0.21 | 17.88±0.11 | 18.06±0.15  | 169.42±2.80 (min) | 261 |
>         | MWT | 23.29±0.57  | 21.56±0.67 | 21.78±0.65 | 168.0±1.27 (min)  | 261 |
>         | **Feat-bias (Ours)** | **78.51±0.16** | **78.71±0.13** | **78.46 ±0.15**  | **145.38 ±1.65 (sec)** | **151** |
>     - **Additional Experiment on STL10 Datasets**. We have previously conducted INR classification tasks on all datasets used in prior works [1][2]. To further underscore the generalization ability of Feat-Bias, we have extended our evaluation to include an INR classification task on the STL10 Datasets. Similar to the case of CIFAR-100, the absence of code implementation in Implicit-Zoo [2] has led us to compare our method solely with the recent state-of-the-art MTL [1]. The result of this experiment are presented below:
>
>
>         |  | Accuracy ↑ | Precision ↑ | F1 ↑ | Time ↓ | Parmas ↓ |
>         | --- | --- | --- | --- | --- | --- |
>         | MWT | 37.22±0.80 | 36.29±0.85 | 35.93±0.23 | 25.03±1.88 (min)  | 261 |
>         | **Feat-bias (Ours)** | **97.62±0.03** | **97.61±0.03** | **97.62 ±0.03**  | **3.84±0.11 (min)** | **151** |
> - **More Formal Mathematical Characterization for How Bias Terms Influence Symmetries of INRs**
>    - We appreciate the valuable suggestion provided by the reviewer, and we hereby present a more formal mathematical formulation. Consider a typical $L$-layer periodically activated representation network, where the output value of the $l$-th layer  is expressed as $\mathbf{z} _ l= f_l(\mathbf{z} _ {l-1}) =  \sin(\omega_0\alpha^l(\mathbf{W} _ l \mathbf{z} _ {l-1}+\mathbf{b} _ l)$. Here, $\mathbf{W} _ l$ and $\mathbf{b} _ l$ denote the weights and bias at layer $l$, $\omega_0$ controls controls the network frequency, and $ \alpha^l$ determines the activation function's periodicity. Specifically, when $l=1$, the activated value of the input layer can be defined as $\mathbf{z} _ 1= f_1(\mathbf{z} _ {0}) = f_1(\tilde{\mathbf{x}} ) = \sin(\omega_0\alpha^1(\sum _ {j=1} ^ {|\mathbf{x}|}\mathbf{W}_ {j} \tilde{\mathbf{x}} _ {j}) +\mathbf{b} _ 1))$, with $\mathbf{x}$ denoting the input coordinates and $\tilde{\mathbf{x}} \in [-1,1] $ represents normalized coordinates( $\tilde{\mathbf{x}} = \frac{2\mathbf{x}}{\max(\mathbf{x}) - \min(\mathbf{x})} - \frac{1}{2}$). The examination of centrally symmetric coordinate pairs $ \langle \tilde{\mathbf{x}} ^ {+} , \tilde{\mathbf{x}} ^ {-} \rangle$, where  $\tilde{\mathbf{x}} ^ {+} = \{ x _ 1, … , x_k \}$ and   $\tilde{\mathbf{x}} ^ {-} = \{ - x _ 1, … , - x_k \}$ (with $k$ denoting the coordinate dimension, e.g., $k=2$ for images), reveals that the presence of $\mathbf{b} _ 1$ disrupts the symmetry effect of the input layer's activated value, resulting in $f_1(\tilde{\mathbf{x}} ^ +) = -f_1(\tilde{\mathbf{x}} ^ -)$. For Bias-free INRs, where $\mathbf{b} _ l = \mathbf{0}, \forall l \in L$, the absence of bias terms leads to $f(\tilde{\mathbf{x}} ^ +) = \sin(\omega_0\alpha^1(\sum _ {j=1} ^ {|\mathbf{x}|}\mathbf{W}_ {j} \tilde{\mathbf{x}} ^ + _ {j}) )) = - \sin(\omega_0\alpha^1(\sum _ {j=1} ^ {|\mathbf{x}|}\mathbf{W}_ {j} \tilde{\mathbf{x}} ^ - _ {j}) )) = -f(\tilde{\mathbf{x}} ^ -)
> $. This leads to a symmetric effect on the output value, where $| \mathbf{y} ^ + |= |\prod _ {l=1} ^ L f_l(\tilde{\mathbf{x}} ^ +)| = |\prod _ {l=1} ^ L f_l(\tilde{\mathbf{x}} ^ -)| = | \mathbf{y} ^ - |$ , which hinders the reconstruction of the signal with spatial aliasing, as natural signals should exhibit  $|\mathbf{{y}}^ + _ {gt} | \neq |\mathbf{{y}}^ - _ {gt} |$.  The presence of bias terms solely in the first layer, i.e.,$ \mathbf{b} _ 1 \neq \mathbf{0}$  ensures that  $|f_1(\tilde{\mathbf{x}} ^ +)| \neq |f_1(\tilde{\mathbf{x}} ^ -)|$ thereby averting failed reconstruction with spatial aliasing.
>
> - **Concerns about Limited Ablation Study**
>     - **The Empirical Studies (Sec.2) as Essential Ablation Study.** The primary focus of the paper is to unveil the previously overlooked significance of bias terms in INRs via empirical ablation studies. Comprehensive ablation studies have been conducted and presented in Fig.2, Tab.2, Tab.4 (Appendix), and Tab.5 (Appendix) to address the question of the impact of bias terms from different layers on INRs. Given this perspective, our proposed approach of exclusively freezing bias terms in input layers does **NOT** necessitate additional ablation studies, as our method does not constitute a pipeline consisting of multiple function blocks.
>     - **The Role of Sec. 4.3 Ablation Study: Supplementary Verification for Spatial Aliasing.** In Sec. 4.3, another ablation study was carried out to confirm the source of spatial aliasing, rather than assessing the efficacy of our method. We believe that our experimental setup adequately confirms the source of spatial aliasing. Additional ablation studies on different datasets can be located in Sec. C of the appendix.
>     - **Ablation Study on Feature Downsampling Method**. Upon careful consideration, we have identified the only suitable area for an additional ablation study, which pertains to the feature downsampling method utilized in Feat-Bia. The results of this ablation experiment have been demonstrated in the Response to **Reviewer x6DW**, presented under the bolded title **Additional Experiments for Feature Sampling**. These results demonstrate the effectiveness of selecting linear interpolation as our method for dimension reduction.
>
> [1] End-to-end implicit neural representations for classification. CVPR, 2025.
>
> [2] Implicit zoo: A large-scale dataset of neural implicit functions for 2d images and 3d scenes. In NeurIPS, 2024.

---

> > ### Comment · Reviewer_sknb · 2025-08-05
> >
> > Thanks to the author's efforts during the rebuttal, I think most of my concerns have been resolved, so I decided to increase my score.

---

> ### Author Response · Authors · 2025-08-05
>
> Dear Reviewer sknb,
>
> We sincerely thank you for your constructive feedback and for acknowledging our work with the improved score. We will incorporate the clarifications and formulations mentioned above into the final manuscript.
>
> Best regards,
>
> The Authors

---

### Official Review · Reviewer_x6DW · 2025-07-03

**Clarity:** 4
**Significance:** 3
**Originality:** 3
**Rating:** 5
**Confidence:** 4

**Summary:**

- The authors propose Feat-Bias which initializes input-layer bias terms of an INR network with features extracted from pretrained models and keeps them frozen during training.
- Feat-Bias results in improved INR classification performance.
- The authors provide an explanation for why the bias terms can be replaced and how that can result in improved performance

**Questions:**

- Why do the authors think that the transferred bias terms result in improved performance? Could it be regularization?

**Ethical Concerns:**

["NO or VERY MINOR ethics concerns only"]

**Final Justification:**

The authors have answered my questions and I have decided to maintain my score of accept.

**Limitations:**

Yes

**Quality:**

3

**Strengths And Weaknesses:**

Strengths:
- The authors provide a clear explanation of how standard bias term training does not improve performance, that the role of bias terms is to prevent spatial aliasing, and that using bias terms in only the input layer is sufficient.
- This is a simple, easy to implement idea that has a performance benefit.
Weaknesses:
- It is not clear to me that knowledge is really being distilled from the pretrained network into the INR bias weights rather than that the pretrained weight initialization has some kind of regularization effect. The fact that significant dimensionality reduction is needed to squeeze the pretrained weights into the INR weights suggests to me that it might be mostly regularization. I would like to have seen a baseline where a distribution was fit to the pretrained weights and then the INR bias weights were randomly sampled from that distribution.

---

> ### Author Rebuttal · Authors · 2025-07-30
>
> We thank the reviewer for your positive feedback and insightful comments. We are delighted by the reviewer's appreciation of the originality, clarity, and effectiveness of our work. Herein, we address the raised concerns as follows:
>
> - **The Reason Why Transferred Bias Terms Result in Improved Performance**. As explained in lines 202 to 220 (Sec. 3), we believe that the Feat-bias can significantly improve INR classification performance because the fixed bias terms serve as ideal storage spaces for high-level features extracted from well-established pre-trained encoders (e.g., ViT or DINOv2). These bias terms act as a kind of distillation, transferring high-level features from large pre-trained networks to smaller input-layer bias spaces. In this manner, each INR can benefit from the representation capabilities of a large pre-trained network, leading to significant improvements when implementing additional networks for downstream INR tasks.
> - **Regularization Effect.** This is a very inspiring assumption. However, we do not think the performance boost comes from dimensionality reduction regularization. The dimension reduction in Feat-Bias can be seen as an operation to address the dimensional mismatch between features from pre-trained encoders and the flexible dimension settings of INRs. For instance, the decoded feature from ViT-base has 768 dimensions, but the number of first-layer biases is determined by the INR size. If the setting is 3x64, we only have 64 dimensions to represent the pre-trained features. In this case, dimension reduction is necessary. Such a dimensionality reduction operation results in some information loss from the pre-trained features, leading to a slight decrease in performance, which can be verified by the following experiment conducted on CIFAR-10:
>
>
>     |  | Accuracy | Precision | F1 |
>     | --- | --- | --- | --- |
>     | MWT (best) [2] | 46.94 | 46.48 | 46.51 |
>     | Scale-GMN-B (best) [1] | 56.14 | 57.67 | 56.12 |
>     | Zoo (best) [3] | 81.57 | 81.53 | 81.51 |
>     | Feat-Bias [768]  | 96.62 / 97.73 | 96.63 / 97.73 | 96.62 / 97.73 |
>     | Feat-Bias [768 → 256]  | 95.99 / 97.05 | 96.01 / 97.06 | 95.99 / 97.05 |
>     | Feat-Bias [768 → 64]  | 91.68 / 93.78 | 91.70 / 93.38 | 91.68 / 93.78 |
>     | Feat-Bias [768 → 32] | 85.98 / 89.44 | 85.94 / 89.44 | 85.94 / 89.44 |
>
>     The experimental settings in this table are strictly aligned with those in Tab. 2 of our paper (lines 245 to 253). The reported results in the row of Feat-bias is with the format of (ViT/DinoV2). As shown, the 768-dimensional configuration demonstrates the best classification performance, indicating that dimensionality reduction negatively affects INR post-processing tasks due to the information loss from the pre-trained encoders. However, we found that by selecting a simple yet appropriate dimensionality reduction method (in our case, linear interpolation), the performance loss remains acceptable, far surpassing existing INR post-processing methods. This trade-off proves to be worthwhile, as it reduces the size of the INR architecture while preserving the memory efficiency advantages inherent to INR.
>
> - **Additional Experiments for Feature Sampling.** We have conducted the sampling-based dimensionality reduction method you proposed and compared it with others. First, directly sampling features from the fittest distribution would break the inherent permutation of encoded features, resulting in suboptimal performance. This is quite similar to random sampling ( $x \sim U(-\frac{1}{\sqrt{n}}, \frac{1}{\sqrt{n}})$), completely losing the prior of the pre-trained encoder. To address this problem, we maintain an additional map to save the permutation of the original features and permute the sampled bias terms according to this map. The experimental results are as follows：
>
>
>     |  | Random | ~ Fittest Dist. | Max pooling | Avg Pooling | Linear (Ours) | ~ Fittest Dist. + Re-permute |
>     | --- | --- | --- | --- | --- | --- | --- |
>     | ViT | 0.1066 | 0.1329 | 0.8288 | 0.8852 | 0.9168 | 0.9027 |
>     | DinoV2 | 0.1121 | 0.1297 | 0.7728 | 0.9185 | 0.9378 | 0.9193 |
>
>     All experiments were conducted under the same settings, aiming to reduce the dimension from 768 to 64 (aligned with the INR classification benchmark in Table 2 of the paper). We tested both ViT and DinoV2 features respectively. The reported values represent classification accuracy on the CIFAR-10 dataset. Among the baselines, *Linear* represents the common linear interpolation method we implemented, and *~ Fittest Dist. + Re-permute* is our re-implementation for feature sampling, as described above. The results show that *~ Fittest Dist. + Re-permute* achieves comparable performance but slightly lags behind our method, while outperforming other baselines. We greatly appreciate the reviewer's academic acumen and will incorporate this baseline into the experimental section of the final manuscript.
>
>
> [1] Scale equivariant graph metanetworks. In NeurIPS, 2024.
>
> [2] End-to-end implicit neural representations for classification. CVPR, 2025.
>
> [3] Implicit zoo: A large-scale dataset of neural implicit functions for 2d images and 3d scenes. In NeurIPS, 2024.

---

> > ### Comment · Reviewer_x6DW · 2025-08-09
> > **rebuttal**
> >
> > Thank you for the additional regularization experiment results. I believe including them in the final paper will strengthen the paper. I will keep my rating of accept.

---

### Comment · Area_Chair_pudH · 2025-08-06

Dear reviewers and authors,

The discussion period is ending soon, so please make sure to finalize the discussion (and make mandatory acknowledgments, if not completed yet). Thank you so much for your active participation!

Best,
AC.

---

### Note · Authors · 2025-08-16

Dear Reviewers and AC,

We extend our sincere gratitude to the reviewers and AC for their valuable comments and suggestions.

In this paper, we challenge the conventional approach of implementing full-bias INR architecture and examine the role of bias terms through intuitive analysis and empirical evidence. We illustrate that bias terms in INRs primarily function to eliminate spatial aliasing induced by symmetry in both coordinates and activation functions, with input-layer bias terms yielding the most substantial benefits. Additionally, we introduce Feat-Bias, which initializes input-layer bias with high-level features to effectively address the performance limitations in INR post-processing tasks, achieving superior accuracy, reducing parameter count, and enhancing reconstruction quality.

During the rebuttal period, we are gratified by the reviewers' recognition of our work as novel (*sknb, 3n6q, aC5b*), effective (*x6DW, aC5b*), clear (*x6DW, 3n6q, aC5b*), and well-motivated (*sknb, 3n6q*). In response to the reviewers' concerns, we provided detailed responses. We believe that all concerns have been adequately addressed during the discussion period, and we deeply appreciate the active participation of the reviewers in the discussion, which has offered additional insights and valuable suggestions for our work. Consequently, all reviewers have provided positive feedback, with three of them (*sknb, 3n6q, aC5b*) acknowledging the contributions of our work by improving the score and one reviewer (*x6DW*) maintaining the rating of acceptance.

We will carefully consider all suggestions raised during the discussion period and will integrate additional insights and explanations into the final version. We express our gratitude to all reviewers and AC for their dedication and time invested in this review process!

Best regards,

The Authors

---

### Decision · Program_Chairs · 2025-09-17

**Decision:**

Accept (poster)

**Comment:**

This paper challenges the conventional full-bias design in implicit neural representations, showing that bias terms mainly mitigate spatial aliasing rather than enhance expressivity. The proposed Feat-Bias method leverages pretrained features as input-layer biases, yielding strong improvements in INR classification and reconstruction efficiency. Reviewers found the idea simple, new, well-substantiated, and practical.

Concerns have been raised regarding limited theoretical understanding and experimental scope. The concerns were largely addressed during the response phase, which led reviewer to be unanimously supportive toward acceptance of the paper. Overall, this is a novel and impactful contribution, and I recommend acceptance.